



# Meso- to micro-scale modeling of atmospheric stability effects on wind turbine wake behavior in complex terrain

Adam S Wise[1], James M T Neher[1], Robert S Arthur[2], Jeffrey D Mirocha[2], Julie K Lundquist[3,4], and Fotini K Chow[1]

[1]Department of Civil and Environmental Engineering, University of California, Berkeley, Berkeley, California, USA
[2]Lawrence Livermore National Laboratory
[3]Department of Atmospheric and Oceanic Sciences, University of Colorado Boulder, Boulder, Colorado, USA
[4]National Renewable Energy Laboratory, Golden, Colorado, USA

**Correspondence:** Adam S Wise (adamwise@berkeley.edu)

**Abstract.** Most detailed modeling and simulation studies of wind turbine wakes have considered flat terrain scenarios. Wind turbines, however, are commonly sited in mountainous or hilly terrain to take advantage of accelerating flow over ridgelines. In addition to topographic acceleration, other turbulent flow phenomena commonly occur in complex terrain, and often depend upon the thermal stratification of the atmospheric boundary layer. Enhanced understanding of wind turbine wake interaction with these terrain-induced flow phenomena can significantly improve wind farm siting, optimization, and control. In this study, we simulate conditions observed during the Perdigão field campaign in 2017, consisting of flow over two parallel ridges with a wind turbine located on top of one of the ridges. We use the Weather Research and Forecasting model (WRF) nested down to micro-scale large-eddy simulation (LES) at 10 m resolution, with a generalized actuator disk (GAD) wind turbine parameterization to simulate turbine wakes. Two case studies are selected, a stable case where a mountain wave occurs and a convective case where a recirculation zone forms in the lee of the ridge with the turbine. The WRF-LES-GAD model is validated against data from meteorological towers, soundings, and a tethered lifting system, showing good agreement for both cases. Comparisons with scanning Doppler lidar data for the stable case show that the overall characteristics of the mountain wave are well-captured, although the wind speed is underestimated. For the convective case, the size of the recirculation zone within the valley shows good agreement. The wind turbine wake behavior shows dependence on atmospheric stability, with different amounts of vertical deflection from the terrain and persistence downstream for the stable and convective conditions. For the stable case, the wake follows the terrain along with the mountain wave and deflects downwards by nearly 100 m below hub-height at four rotor diameters downstream. For the convective case, the wake deflects above the recirculation zone over 50 m above hub-height at the same downstream distance. This study demonstrates the ability of the WRF-LES-GAD model to capture the expected behavior of wind turbine wakes in regions of complex terrain, and thereby to potentially improve wind turbine siting and operation in hilly landscapes.



## 1 Introduction

Wind turbines are commonly sited in mountainous and hilly complex terrain to take advantage of topographic flow enhancement, such as the acceleration of flow over ridge lines and hill tops. The complexity of the interaction between wind turbine wakes and complex terrain has historically limited research studies to wind tunnel experiments and numerical simulations with simple Gaussian and sinusoidal hills (see Porté-Agel et al. (2020) for a recent review on wind farm flows and topography). In addition to topographic acceleration, other terrain-induced flow phenomena in the atmospheric boundary layer (ABL) may affect wind turbine performance and wake propagation and characteristics (Xia et al., 2021; Draxl et al., 2021). These microscale processes include mountain/lee waves (and associated rotors), hydraulic jumps, valley flows, flow separation and recirculation (Fernando et al., 2019; Baines, 1998). The dynamics of terrain-induced flow phenomena can vary widely based on the time of the day, with lee waves only occurring during stably stratified conditions, typical at night, and flow separation or recirculation often occurring during daytime convective conditions.

Many studies have demonstrated how wind turbine wake behavior is strongly affected by ABL stability conditions (Iungo et al., 2013; Mirocha et al., 2014; Aitken et al., 2014; Bhaganagar and Debnath, 2014; Abkar and Porté-Agel, 2014; Bhaganagar and Debnath, 2015; Dörenkämper et al., 2015; Abkar et al., 2016; Vollmer et al., 2017; Bodini et al., 2017; Englberger and Dörnbrack, 2018). The wake, as a result of wind turbine operation, induces a decrease in wind speed and increase in turbulence, resulting in lower than expected power output and increased structural loading in downstream turbines (Lissaman, 1979; Taylor, 1990; Thomsen and Sørensen, 1999; Frandsen and Christensen, 1994; Frandsen and Thomsen, 1997; Dahlberg et al., 1992). During unstable or convective conditions (typically during the daytime for onshore locations), wakes diffuse rapidly and meander due to large-scale eddies that are present. During stable conditions, wake deficits can persist for much greater downstream distances (Hirth and Schroeder, 2013).

Large-eddy simulation (LES) is a common and useful tool for simulating the ABL. LES explicitly solves for the most energetic eddies while parameterizing the effects of the smaller length scales on the resolved-scale flow. This allows for resolving transient turbulent structures, which are important features of boundary layer turbulence and wind turbine wakes. LES is increasingly complementing and even in many cases replacing lower-fidelity techniques as a means to investigate wind turbine wake effects (see Stevens and Meneveau (2017) for a recent review). One example is Shamsoddin and Porté-Agel (2017), who modeled a wind farm sited on a single hill and validated their model using data from Tian et al. (2013) for a neutral ABL.

In the present work, we utilize the LES capability of the Weather Research and Forecasting (WRF) model (Skamarock et al., 2008; Powers et al., 2017) together with an actuator disk wind turbine model to simulate wind turbine wake behavior in complex hilly terrain. In addition to running standalone, idealized WRF-LES simulations, we take advantage of the more complex multi-scale modeling framework available within WRF. This multi-scale framework in WRF downscales mesoscale forcing to microscale LES using a grid nesting approach, with lateral boundary conditions provided from each nest's parent domain. Such setups can provide LES with more realistic time-varying inflow conditions directly from the mesoscale simulations, capturing a wider range of atmospheric phenomena than conventional idealized LES setups.



Moreover, implementation of a wind turbine actuator disk parameterization within the nested LES domain provides a unique high-fidelity simulation framework for wind turbine wake prediction in turbulent flow settings under more realistic environmental and atmospheric forcing conditions. This study utilizes a generalized actuator disk (GAD) wind turbine parameterization on the finest domain following Mirocha et al. (2014). The GAD tool, hereinafter referred to as WRF-LES-GAD for the model in its entirety, has been previously validated and shown to capture turbine-airflow interactions and wake behavior well at grid
resolutions of 10 m (Mirocha et al., 2015; Aitken et al., 2014; Marjanovic et al., 2017; Arthur et al., 2020).

The test location chosen in this study is that of the Perdigão field campaign (Fernando et al., 2019), which took place in 2017 in Portugal. The Perdigão experiment characterized the flow over two parallel ridges with a wind turbine located on the southwest ridge and provided valuable data for characterizing wind turbine wakes in complex terrain (Menke et al., 2018; Barthelmie and Pryor, 2019; Wildmann et al., 2018, 2019). Menke et al. (2018) classified wind turbine wake behavior based
on atmospheric stability using scanning Doppler lidars at Perdigão. They identified four different cases for the stratification: "stable + mountain wave" where the wake advected downwards following the terrain, "stable" and "neutral" cases where the wake remained at a constant height above sea level, and "unstable" cases where the wake advected upwards. Barthelmie and Pryor (2019) similarly characterized wake behavior at Perdigão based on atmospheric stability. Using measurements averaged over longer time periods (10 minutes compared to 24 seconds), they infer that all wakes were initially lofted and then strongly
influenced by stability, with wake centers moving downwards in unstable conditions and also generally moving downwards but remaining at greater heights during stable conditions. The different findings in these two studies motivate the need to further study wind turbine wakes in complex terrain and to accompany the field data analysis with numerical models.

Two LES modeling studies of the Perdigão campaign have been presented in the literature thus far, both in idealized conditions with neutral stability. Berg et al. (2017) performed LES of the Perdigão site in neutral atmospheric stability conditions
and showed that the steep terrain in Perdigão resulted in the formation of a recirculation zone with which the wake did not interact. Instead, the wake advected at a constant height above sea level like the "neutral" case characterized by Menke et al. (2018). Dar et al. (2019) also modeled the Perdigão site using LES to examine the self-similarity of wind turbine wake behavior as a function of varying terrain complexity under neutral stratification. They found that self-similarity is preserved for a shorter distance compared to what is observed in flat terrain and that the wake propagation was similar to that seen by Berg et al.
(2017). Both of these studies considered idealized conditions with neutral atmospheric stability.

The focus of this work is to model realistic atmospheric conditions and the associated turbulent flow phenomena to better understand wind turbine wake propagation in complex terrain, using the Perdigão site as the test location (described in Sect. 2). Specifically, we analyze how the vertical deflection and dissipation of the wake varies based on atmospheric stability. In our modeling approach, we use the WRF-LES-GAD framework, first in an idealized setting and then in a multi-scale nested
domain setup with real atmospheric forcing (Sect. 3). The predicted flow structure during stable and convective case studies is analyzed and validated by comparing with scanning lidars, meteorological towers, soundings, and a tethered lifting system (Sect. 4). These findings are followed by a detailed analysis of the wind turbine wake behavior in the given cases.

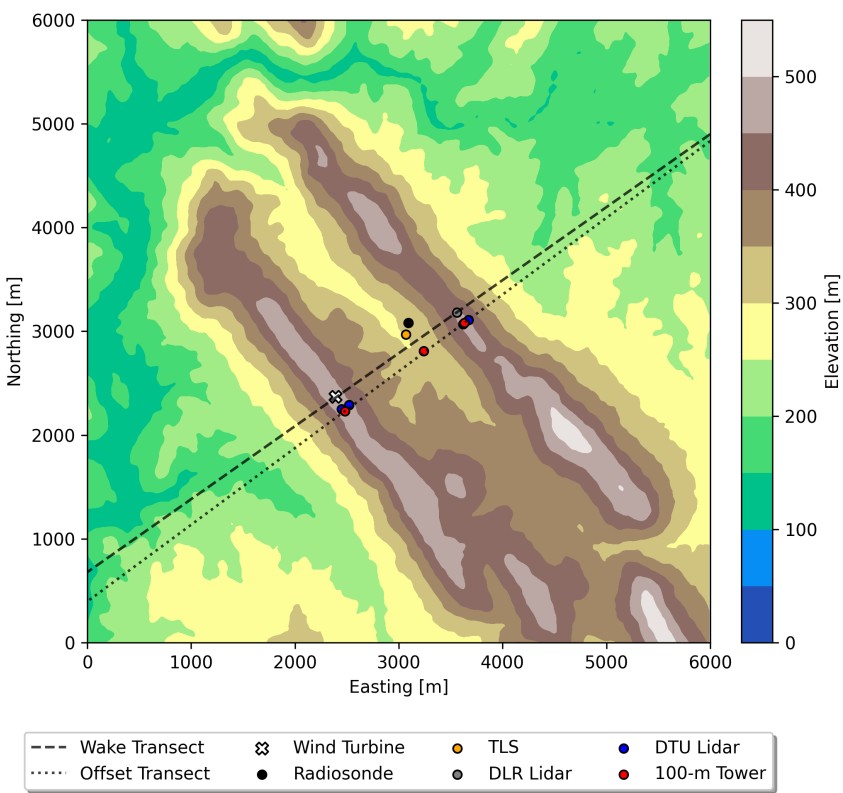

**Figure 1.** Terrain and layout of the Perdigão site showing the location of relevant instrumentation. The Wake and Offset Transects are used for analysis later in this paper.

## 2 Case Study

### 2.1 Overview of the Perdigão campaign

The Perdigão field campaign was a European Union - United States collaboration of over 70 scientists, engineers, and support personnel. The Perdigão site, named for a village located near the Vale do Cobrão in eastern Portugal, consists of two parallel ridges with a 2 megawatt (MW) wind turbine located on the southwest ridge (see Fig. 1). This site was selected because the straight valley extends over 2 km in distance, suggesting that the flow could be representative of an idealized two-dimensional valley flow in nature. Additionally, the annual climatology showed that common wind directions measured at ridge-top are

perpendicular to the ridges (Fernando et al., 2019).

During the intensive operation period from 1 May 2017 to 15 June 2017, a comprehensive, high-resolution dataset was collected. Instrumentation included 100-m meteorological towers (hereinafter referred to as met towers), radiosondes, lidars, a



tethered lifting system (TLS) and additional instrumentation including numerous shorter meteorological towers and radiometers. The instrumentation of interest is discussed here, and the layout of the Perdigão field site is shown in Fig. 1.

Three 100-m met towers were placed along a transect roughly perpendicular to the ridge (see Fig. 1). One tower was located on the southwest ridge (SW_TSE04), another in the valley (V_TSE09), and the third on the northeast ridge (NE_TSE13). SW_TSE04, located approximately 150 m southeast of the wind turbine laterally along the ridgeline, is representative of inflow conditions. The three 100-m met towers have booms angled at 135° from north located at heights of 10, 20, 40, 60, 80, and 100 m with sonic anemometers sampling three-dimensional wind at 20 Hz. In this paper, data representative of near

surface winds (10 m) and hub-height (80 m) are examined. In addition to sonic anemometers, temperature sensors are available at multiple heights. The temperature sensors at 10 m and 100 m are used to characterize atmospheric stability using a simple temperature gradient similar to Menke et al. (2019).

A radiosonde launching site was located in the valley (Fig. 1) and provides vertical profiles of pressure, temperature, wind speed, and wind direction. The sondes were launched every 6 hours at approximately 00:00, 06:00, 12:00, 18:00 UTC for the

duration of the campaign. During periods of interest, additional soundings were launched.

A tethered lifting system (TLS) from the University of Colorado, Boulder, was also launched from the valley. The TLS is a unique device that is able to obtain in-situ measurements of wind speed, wind direction, and temperature aloft at 1 kHz (Balsley, 2008). Additionally, GPS measurements of latitude, longitude, and altitude were sampled at 5 s. When in profiling mode, the TLS ascends and descends at a rate of approximately 0.3 m s$^{-1}$ and covers a vertical range of close to 400 m. The capabilities

of the TLS have been demonstrated in several field campaigns, including a previous wind turbine wake investigation (Lundquist and Bariteau, 2015).

Simulation results are also compared with scanning Doppler lidar data. The first is a scanning lidar operated by the German Aerospace Center (DLR) located on the northeast ridge (Wildmann et al., 2018). This lidar's scanning trajectory is perpendicular to the wind turbine rotor allowing for the wake to be captured in addition to relevant flow features in the valley. The second

set of scanning lidars used here were operated by the Danish Technical University (DTU) (Menke et al., 2020). Four lidars, two on each ridge, scanned a transect perpendicular to the valley. Because of how the lidars are arranged, a multi-Doppler lidar scan of the valley, the two ridges, and the surrounding area can be formed. This multi-Doppler lidar scan does not capture the wind turbine wake but does show microscale features that interact with the wind turbine wake.

## 2.2    Case Studies and Phenomena of Interest

Two terrain-induced flow features are of interest for the present work: (a) mountain waves and (b) recirculation zones. These flow features may occur at the Perdigão site depending on the time of day (Menke et al., 2019; Fernando et al., 2019; Wagner et al., 2019). Mountain waves can occur when stably stratified flow approaches a topographic disturbance, such as a mountain ridge. These waves can be characterized by the Froude number:

$$Fr = \frac{U}{hN} \tag{1}$$



where $h$ is the mountain ridge height, $U$ is the free stream wind speed and $N$ is the Brunt-Väisälä frequency, defined as:

$$N = \sqrt{\frac{g}{\theta}\frac{d\theta}{dz}} \qquad (2)$$

Here $g$ is the gravitational constant, $\theta$ is the potential temperature of the environment, and $\frac{d\theta}{dz}$ is the vertical gradient in potential temperature. For small Froude numbers ($< 1$), when wind speeds are low or the stability is very strong, the wavelength of the mountain wave is shorter than the width of the mountain resulting in weak mountain waves. Mountain waves resonate when

the Froude number is approximately equal to 1, resulting in strong up and downdrafts where rotors (and potentially hydraulic jumps) are present (Jackson et al., 2013). When wind speeds are strong and stability is weak, the Froude number is large ($> 1$) resulting in long wavelengths with the potential for reverse flow in the lee of the mountain. Neutral and convective conditions represent a theoretically infinite Froude number. For Fr $\approx \infty$, waves do not occur; instead, a turbulent mountain wake with recirculation forms. Relevant to this study are Fr $\approx 1$ during stable conditions and Fr $\approx \infty$ during convective conditions. For

more information regarding mountain waves, the reader is referred to (Baines, 1998; Jackson et al., 2013).

We use the potential temperature gradient between 10 m and 100 m on the met tower of the southwest ridge to capture inflow conditions and quantify stability. Note that SW_TSE04 was not outfitted with any pressure sensors; therefore, the potential temperature is approximated by $\theta \approx T + (g/C_p) \cdot z$ where $(g/C_p) = 0.0098$ K m$^{-1}$ as was done by Menke et al. (2019). A positive temperature gradient indicates stably stratified flow and a negative temperature gradient indicates unstable

or convectively driven flow.

### 2.2.1 Case 1: Stable Conditions

The early morning of 14 June 2017 shows typical night-time stably-stratified conditions as indicated by a positive temperature gradient in Fig. 2(e). From 04:00 to 06:00 UTC, the hub-height wind direction is mostly constant and perpendicular to the ridges (Fig. 2(a) and Fig. 2(b), respectively). The wind speed is between 5 and 7.5 m s$^{-1}$, a range in which the turbine will

operate and generate wake effects. Figure 3 shows that the Brunt-Väisälä frequency is close to 0.04 s$^{-1}$ at the beginning of the period decreasing to 0.02 s$^{-1}$ at the end of the period. Using the 80 m wind speed from SW_TSE04, the Froude number begins the period close to 0.7, approaches 1.0 during the middle of the period, and then decreases at the end of the period as the sun begins to rise. During this period, two soundings were launched and the TLS was operational.

### 2.2.2 Case 2: Convective Conditions

Typical daytime convective conditions on 13 May 2017 are indicated by the negative temperature gradient in Fig. 2(f). For much of the day (08:00-16:00 UTC), the temperature gradient is approximately -0.005 K m$^{-1}$. During the period of interest (13:00-14:00 UTC), the flow is moderately convective, the hub-height wind is perpendicular to the ridges, and the hub-height wind speed is mostly constant at 7 m s$^{-1}$ (Fig. 2(d)). Similar to the stable case study, a wind speed of 7 m s$^{-1}$ will cause the wind turbine to operate and generate wake effects. During this period, neither the TLS nor any soundings were launched,

therefore the met towers provide the main metric for quantitative validation.

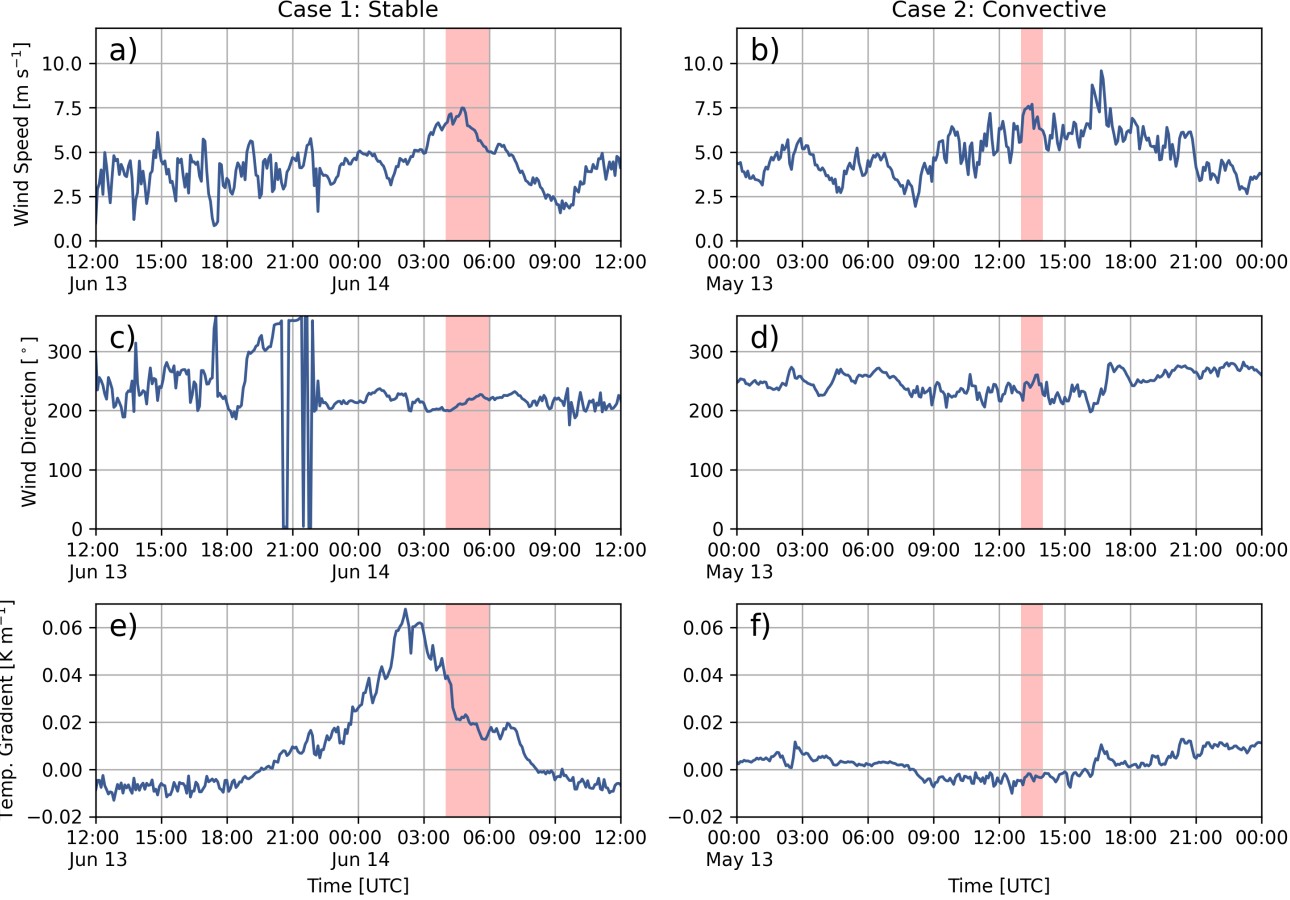

**Figure 2.** Hub-height wind speed, hub-height wind direction, and potential temperature gradient for the stable case and for the convective case. Data are from SW_TSE04 and the periods of interest are highlighted in red. Data have been subsampled to five minute intervals.

## 3 Methods

### 3.1 WRF-LES-GAD

The present work utilizes the large-eddy simulation capability of the WRF model. The base source code is from version 3.7.1, with modifications including vertical grid nesting (Daniels et al., 2016), the generalized actuator disk (GAD) (Mirocha et al., 2014), and a turbine yawing capability (Arthur et al., 2020).

The GAD requires specifications for the turbine's airfoil lift and drag coefficients. The turbine located at the Perdigão site is a 2.0-MW E-82 Enercon turbine; however, the required lift and drag parameters are not publicly available. We therefore use the wind turbine parameterization from Arthur et al. (2020), which is similar but not identical to the turbine at Perdigão. Both



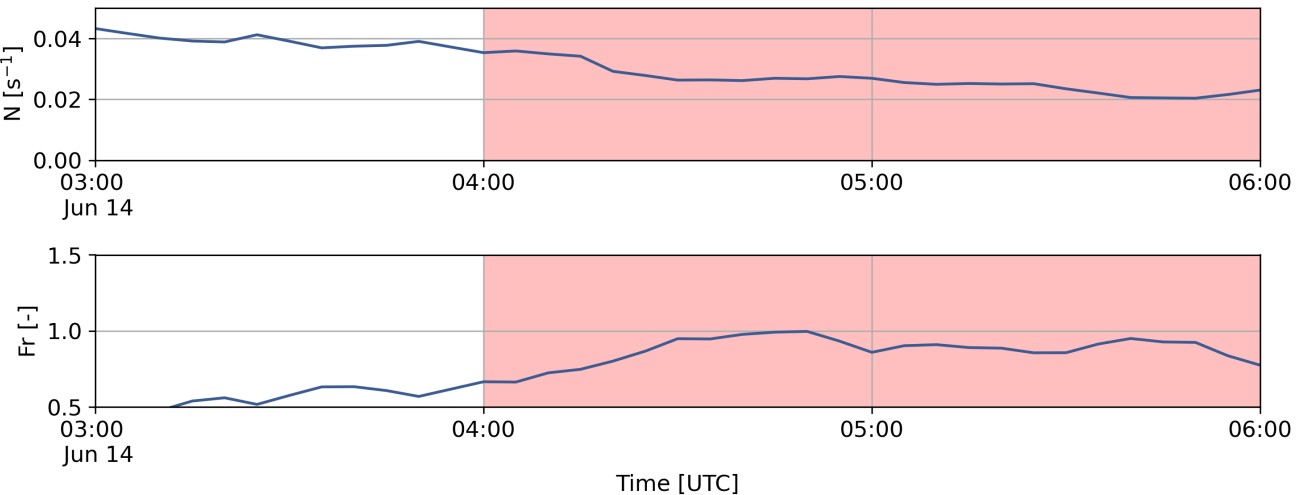

**Figure 3.** Calculated Brunt-Väisälä frequency (N) and Froude number (Fr) during the stable case study. The period of interest is highlighted in red.

turbine types have a roughly 80-m hub-height and rotor diameter, and are therefore expected to create similar wake effects. Minor differences between the two turbines are not expected to be critical to the conclusions of this study.

WRF-LES-GAD can be setup in a standalone LES-only configuration, or in a meso- to micro-scale (i.e., multi-scale) configuration. In the LES-only configuration, the wind speed, wind direction, surface heat flux, and various other environmental parameters can be set by the user. We refer to this as a semi-idealized setup.

### 3.2 Semi-Idealized Simulation Setup

The three-dimensional semi-idealized setup uses a single domain with periodic boundary conditions. The grid resolution is 10 m and the two-dimensional terrain is derived from a cross-section of the Perdigão terrain which intersects the wind turbine and is perpendicular to the ridges (Wake Transect in Fig. 1). The two-dimensional terrain extends 2500 m in the span-wise direction and 5000 m in the streamwise direction. We conducted two semi-idealized simulations, one to imitate the stable case study and the other to imitate the convective case study.

For the stable semi-idealized simulation, the sounding launched on 14 June 2017 at 03:55 UTC in the valley at Perdigão is used for initialization. The atmospheric boundary layer in the sounding is stably stratified with a low-level jet present. Short-term simulations (Fig. 4(a)) capture the jet deforming into a mountain wave after the first (southwest) ridge. Longer-term simulations are not possible as the stratification is self-destructive with periodic boundary conditions, which highlights the limitations of this semi-idealized setup.

For the convective semi-idealized simulation, we use a sounding with constant (7 m s$^{-1}$) wind speed and potential temperature profiles. Convective conditions are prescribed using a uniform surface heat flux of 100 W m$^{-2}$. Initial simulations





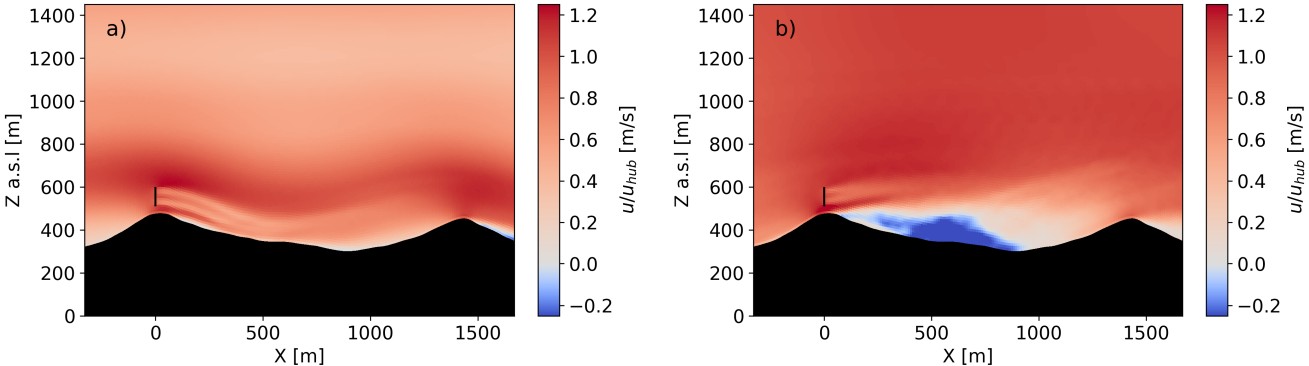

**Figure 4.** Streamwise velocity ($u$) normalized by the hub-height streamwise velocity ($u_{hub}$) for (a) the stable and (b) the convective semi-idealized setups.

(not shown) showed that recirculation in the valley did not occur (Wise et al., 2020). Further investigation determined that recirculation did not occur due to the small surface roughness length, $z_0$, specified in the setup. Initially, we used $z_0 = 0.05$, which corresponds to the land use classification for much of the Perdigão site. The CORINE Land Cover data (Bossard et al., 2000) classifies much of the area as mixed shrubland/grassland which is transformed into United States Geological Survey (USGS) land use types to obtain surface roughness lengths (Pineda et al., 2004). Mixed shrubland/grassland equates to surface roughness lengths of 0.01 - 0.05 m (depending on the season). After setting the surface roughness length to 0.5 m, which is reasonable given the tall eucalyptus and fir trees covering most of the valley, recirculation, as defined by flow in the negative X-direction, did occur in the convective semi-idealized simulation (see Fig. 4(b)). Wagner et al. (2019) similarly concluded that the surface roughness lengths at the Perdigão site based on the CORINE Land Cover data were too small.

### 3.3 Multi-Scale Simulation Setup

Having demonstrated the ability of WRF-LES-GAD to capture the dynamics of interest in a semi-idealized setup, we now proceed to the full multi-scale simulation. A 5-domain, nested multi-scale setup for WRF-LES-GAD (see Table 1) captures both mesoscale forcing and microscale features. The outermost domain, d01, has a horizontal grid resolution of 6.75 km, and the innermost domain, d05, has a horizontal grid resolution of 10 m. The two coarsest domains are mesoscale simulations that use a planetary boundary layer (PBL) scheme, while the three inner domains use a microscale large-eddy simulation turbulence closure. There is a relatively large parent grid ratio of 15 from d02 to d03, intentionally chosen to skip across the gray zone where turbulence is only partially resolved (Wyngaard, 2004; Chow et al., 2019; Haupt et al., 2019; Muñoz-Esparza et al., 2017). The multi-scale setup also makes use of vertical grid nesting (Daniels et al., 2016) to adjust the vertical resolution of each domain, with the grid spacing is as fine as 8 m near the surface for d05 while increasing the grid spacing for the coarser



**Table 1.** Parameters used for the nested multi-scale WRF-LES-GAD setup. For the vertical resolution, $\Delta z_{min}$ is for the first grid point above the surface and is approximate due to the nature of the terrain-following coordinate system in WRF.

| Domain | $\Delta x$ [m] | Nest ratio | $\sim\Delta z_{min}$ [m] | $N_x \times N_y$ | $\Delta t$ [s] | turb. closure |
|--------|------|------------|------------|-----------|---------|---------------|
| d01 | 6750 | - | 60 | $141 \times 141$ | 30 | 2.5-level MYNN |
| d02 | 2250 | 3 | 60 | $181 \times 181$ | 10 | 2.5-level MYNN |
| d03 | 150 | 15 | 30 | $271 \times 271$ | 0.5 | TKE 1.5 |
| d04 | 50 | 3 | 30 | $271 \times 271$ | 0.0833 | TKE 1.5 |
| d05 | 10 | 5 | 8 | $601 \times 601$ | 0.0167 | TKE 1.5 |

domains, up to roughly 60 m near the surface for d01. Additionally, most of the vertical resolution is devoted to the bottom ~2 km of the atmosphere, above which there is a sharp drop off in resolution. The coarser vertical resolution for d01 - d04 has the benefit of reducing computational costs as well as maintaining an aspect ratio ($\Delta x/\Delta z$) closer to unity, which is desired for large-eddy simulation (Daniels et al., 2016). Because of the steepness of the terrain, numerical stability requires the use of
very small time steps (as small as 1/60 s for d05). Domains d01, d02, and d03 were spun up for 9 hours to allow for adequate turbulence to develop, prior to starting d04, and d05. An overview of the nested-domain setup showing the geographical extent and topography of each of the domains is shown in Fig. 5.

All three LES domains (d03, d04, and d05) use a stochastic inflow perturbation method known as the cell perturbation method (CPM) to improve the downscaling of coherent, turbulent structures in the nested approach (Muñoz-Esparza et al.,
2014, 2015) in a range of stability conditions (Muñoz-Esparza and Kosović, 2018). The CPM works by applying small temperature perturbations to the flow on the upwind domain boundaries, at essentially no additional computational cost. While the use of high-resolution terrain data is expected to spur turbulence, Connolly et al. (2021) found that during convective conditions in the Perdigão valley, using the CPM further improves the representation of turbulence. Additionally, they found that the CPM improves the rate at which smaller-scale turbulence forms, with no known negative impacts on the flow from the perturbations.
Similar findings regarding CPM are described by Arthur et al. (2020) for nested WRF-LES-GAD simulations of a wind farm in less complex terrain. CPM therefore allows us to reduce the spin-up time and fetch required for the high-resolution domains to develop realistic turbulent structures.

WRF-LES-GAD uses a third-order Runge-Kutta time advancement scheme, with fifth-order horizontal and third-order vertical advection schemes. Relevant physical parameterizations selected include the Eta (Ferrier) scheme for microphysics (Rogers
et al., 2001), the Noah land surface model (Chen and Dudhia, 2001), the Rapid Radiative Transfer Model for longwave radiation (Mlawer et al., 1997), and the Dudhia shortwave radiation model (Dudhia, 1989). The mesoscale simulations, d01 and d02, use the Mellor-Yamada–Nakanishi–Niino (MYNN) level-2.5 PBL scheme (Nakanishi and Niino, 2006, 2009), and d03, d04, and d05 use the turbulent kinetic energy (TKE) level-1.5 LES closure (Deardorff, 1980). All domains use the MYNN surface layer scheme (Nakanishi and Niino, 2006, 2009).





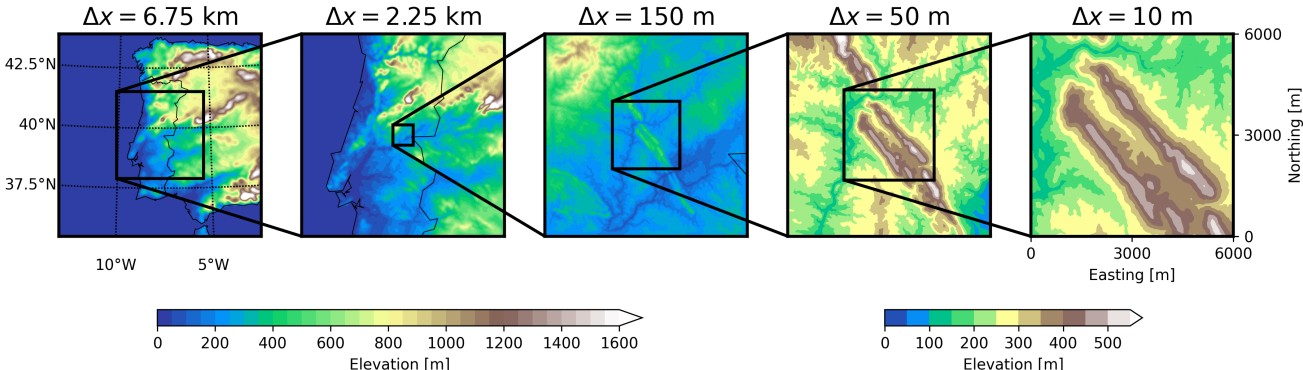

**Figure 5.** Topography of domains used in the multi-scale simulation centered over the Perdigão site. The five domains have resolutions of 6.75 km, 2.25 km, 150 m, 50 m, and 10 m. Dimensions of each domain and other configuration information are included in Table 1.

Initial and boundary conditions for the outermost domain are obtained from reanalysis data. We use Global Forecast System (GFS) data from the National Center for Environmental Protection (National Centers for Environmental Prediction, National Weather Service, NOAA, U.S. Department of Commerce, 2015), made publicly available by the National Oceanic and Atmospheric Administration. In a sensitivity study comparing both GFS and European Centre for Medium-Range Weather Forecasts data for the boundary conditions, the GFS data produced more accurate results for the dates of interest here (Wendels, 2019).

Land cover data was obtained from the Coordination of Information on the Environment (CORINE) at a resolution of 100 m, much finer than the default land cover data provided by NCAR for use in WRF. The CORINE Land Cover 2006 raster dataset (Bossard et al., 2000) is transformed into United States Geological Survey (USGS) land use types to obtain surface roughness lengths for WRF (Pineda et al., 2004), while the surface roughness length of the mixed shrubland/grassland land use index is manually set to 0.5 m. Additionally, high-resolution terrain data (1-arc-second, approximately 30 m) were obtained from the Shuttle Radar Topography Mission (Farr et al., 2007). This high resolution terrain data is required to resolve flow features within the narrow valley.

## 4 Results and Discussion

### 4.1 Stable Case Study

The stable case selected on 14 June 2017, 04:00 - 06:00 UTC is influenced by a mountain wave event. To understand the scale of the mountain event, we can compare the model with multi-Doppler lidar scans obtained by the DTU lidars. The lidars measure line-of-sight or radial velocities, so we project the $u$- and $v$-velocities from the model onto a rotated coordinate system aligned with the Wake Transect in Fig. 1 for comparison. We use this transect to illustrate the wind turbine wake behavior. Note that this transect is oriented approximately 48° from north. The lidars, however, are oriented along the Offset Transect in Fig. 1, which is oriented approximately 49° from north. For the lidar, the line-of-sight velocities have been converted into just the





horizontal component of the velocity. This results in a comparison of along-transect horizontal velocities between the model and the lidars shown in Fig. 6.

The model predicts the wavelength of the mountain wave to be 1200 m (Fig. 6(a) and Figure 6(c)), which is slightly shorter than the 1500 m ridge-to-ridge distance at Perdigão. This prediction by the model is a slight underestimate of the measurements captured by the DTU lidars, where the wavelength is closer to 1400 m; however, qualitatively, the model captures the mountain

wave and associated turbulence quite well. The transects shown in Fig. 6 are nearly 7 km in length which is almost the entire extent of the computational domain. The flow follows the terrain over each ridge, creating a small rotor in the Perdigão valley and a larger rotor beneath the third wave crest downstream of the valley. Qualitatively, in both the model and multi-Doppler lidar scans, turbulent eddies are visible in the rotors with very little turbulence in the wave itself.

In Fig. 6(a), the model also shows a small recirculation zone in the lee of the second ridge ($x \approx 1700$ m) which is not apparent

in the multi-Doppler lidar scans of Fig. 6(b); however, just a few minutes later in Fig. 6(c) and Fig. 6(d), this recirculation zone is apparent in the multi-Doppler lidar scans but not the model. These differences highlight the intermittent and dynamic nature of the phenomena of interest, even though the wind speed and wavelength of the mountain wave are relatively constant with time. Given the complexity of the flow, the overall qualitative agreement between the model and the lidar observations is quite remarkable.

The flow structure of the mountain wave captured by the model also agrees well with measurements from the TLS and met towers. Figure 7 shows the instantaneous wind speed from the model overlaid with measurements from the TLS, and met towers V_TSE09 and NE_TSE13, at times when the TLS is near the surface, halfway up the ascent, near the top of its ascent, and halfway down its descent (the tower on the southwest ridge is omitted for clarity). Additionally, in Fig. 7, a time series from the model, a virtual TLS, is extracted using the GPS position at the corresponding time step of the actual TLS. Comparison

of time-dependent turbulent flow fields from a model and a measurement system that moves in three-dimensions over time is difficult because the instantaneous positioning of turbulent flow features will likely not match. To partially account for this and for any uncertainty in the TLS positioning, time series with an easting and northing position $\pm 30$ m are shown in Fig. 7(e) but with a lighter shading. Additionally, the wind fields in Fig. 7(a-d) have also been spatially averaged by $\pm 30$ m in the span-wise direction. During the two ascents, the virtual TLS and real TLS show good agreement with a root mean squared error (RMSE)

under 2.0 m s$^{-1}$. However, the model slightly underestimates the strength of the jet and there is a negative bias of -0.78 m s$^{-1}$ over the course of the two ascents and descents. Between 04:30 - 04:45 UTC and 05:15 - 05:30 UTC, when the TLS is at its maximum in altitude, the model wind speed is 1-2 m s$^{-1}$ lower than observations. During the descents of the TLS, the model decreases in wind speed slightly sooner, or at a higher altitude, compared to the actual TLS. Given the challenges in such a comparison, the overall agreement between the TLS and the model output is notable.

The 10-m and 80-m data points on the met tower in the valley indicate decreased shear compared to the met tower on the downstream ridge. The model predicts that the 10-m and 80-m locations for V_TSE09 are located below the mountain wave and in a region of more well-mixed and coherent turbulence. The TLS rises above the region of well-mixed turbulence and into the mountain wave. There is a small underestimation in wind speed when the TLS is near the top of its ascent (1 to 2 m s$^{-1}$) but the model otherwise matches measurements well. Near-surface wind speeds on the downstream ridge show some discrepancies



**Figure 6.** Instantaneous along-transect horizontal velocity during the stable case study for (a) the model and (b) the DTU multi-Doppler lidar scan at approximately 04:20 UTC, and for (c) the model and (d) the DTU multi-Doppler lidar scan at approximately 04:40 UTC. The output from the model is instantaneous while the multi-Doppler measurements are over the course of a single scan, which takes about 24 s. The model transects are aligned with the Wake Transect and the lidar transects are aligned with the Offset Transect in Fig. 1. An animation of model results during the entire two-hour period are included in the supplementary material (see Video 1, in the Video Supplement).



**Table 2.** Wind speed, wind direction, and temperature gradient biases and errors between WRF-LES-GAD and meteorological tower measurements.

| Parameter | SW_TSE04 | | V_TSE09 | | NE_TSE13 | |
|---|---|---|---|---|---|---|
| | Bias | RMSE | Bias | RMSE | Bias | RMSE |
| 80 m Wind Speed (m s$^{-1}$) | -0.22 | 0.83 | -0.51 | 2.01 | -0.69 | 1.34 |
| 10 m Wind Speed (m s$^{-1}$) | -0.48 | 0.81 | 0.98 | 1.25 | 2.65 | 2.90 |
| 80 m Wind Direction (°) | -0.4 | 8.4 | -12.7 | 64.1 | 0.0 | 11.3 |
| 10 m Wind Direction (°) | -8.1 | 18.5 | -28.9 | 120.5 | -0.2 | 19.0 |
| Temperature Gradient (K m$^{-1}$) | -0.003 | 0.011 | -0.014 | 0.017 | 0.004 | 0.007 |

(sometimes on the order of 3 to 4 m s$^{-1}$) but the 80 m wind speeds match well throughout the period. Note, however, that exact agreement is not expected in these instantaneous snapshots.

Figure 7 also shows that the wind turbine wake, while time-varying, is deflected by the mountain wave into the valley. The wake likely then meanders out of the transect since the transect and wind direction are slightly misaligned, and also because the wake veers (discussed more in Sect. 4.3). The transects still indicate that the wind turbine wake can propagate at least 800 m 290 downstream, or close to 10 rotor diameters.

Wind speed, wind direction, and the potential temperature gradient are also compared with met tower measurements. Wind speed and wind direction are generally predicted well by the model at both hub-height (80 m) and near the surface (10 m), with greater variability in the valley compared to on top of the two ridges. The 100 m - 10 m potential temperature gradient also compares favorably with measurements but with a longer duration of agreement on top of the two ridges relative to the 295 valley. Figure 8 compares the 80 m wind speed and wind direction and the temperature gradient between the met towers and the model. The model wind speed and direction are output at 10 s intervals while the temperature is output every 150 s. Also note that the three towers lie along the Offset Transect in Fig. 1. Errors quantified in terms of bias and RMSE are shown in Table 2. At hub-height and along the two ridges, wind speed errors are below 1.5 m s$^{-1}$ and wind direction errors are below 12°. The variability in the inflow can be quantified by the standard deviation which is 0.95 m s$^{-1}$ for the model and 0.72 m s$^{-1}$ 300 for SW_TSE04. Within the valley, the wind speed and wind direction agree well with measurements from V_TSE09, but less so compared to the measurements at the top of the ridges, with hub-height wind speed and wind direction errors on the order of 2 m s$^{-1}$ and 60°, respectively. Both the wind speed and wind direction fluctuate much more in the valley compared to the ridges, which the model captures fairly well considering the low wind speeds present. The temperature gradient within the valley in the model is indicative of a well-mixed region whereas the measurements indicate more stable stratification, although 305 this stratification does vary significantly over the period of interest.

The vertical structure of the flow is investigated next by comparing the model with two soundings. The first sounding was launched five minutes prior to the analysis period, but obtained data into the period of interest therefore providing a useful comparison. Figure 9 shows the wind speed, wind direction, and potential temperature for the soundings launched at 03:55



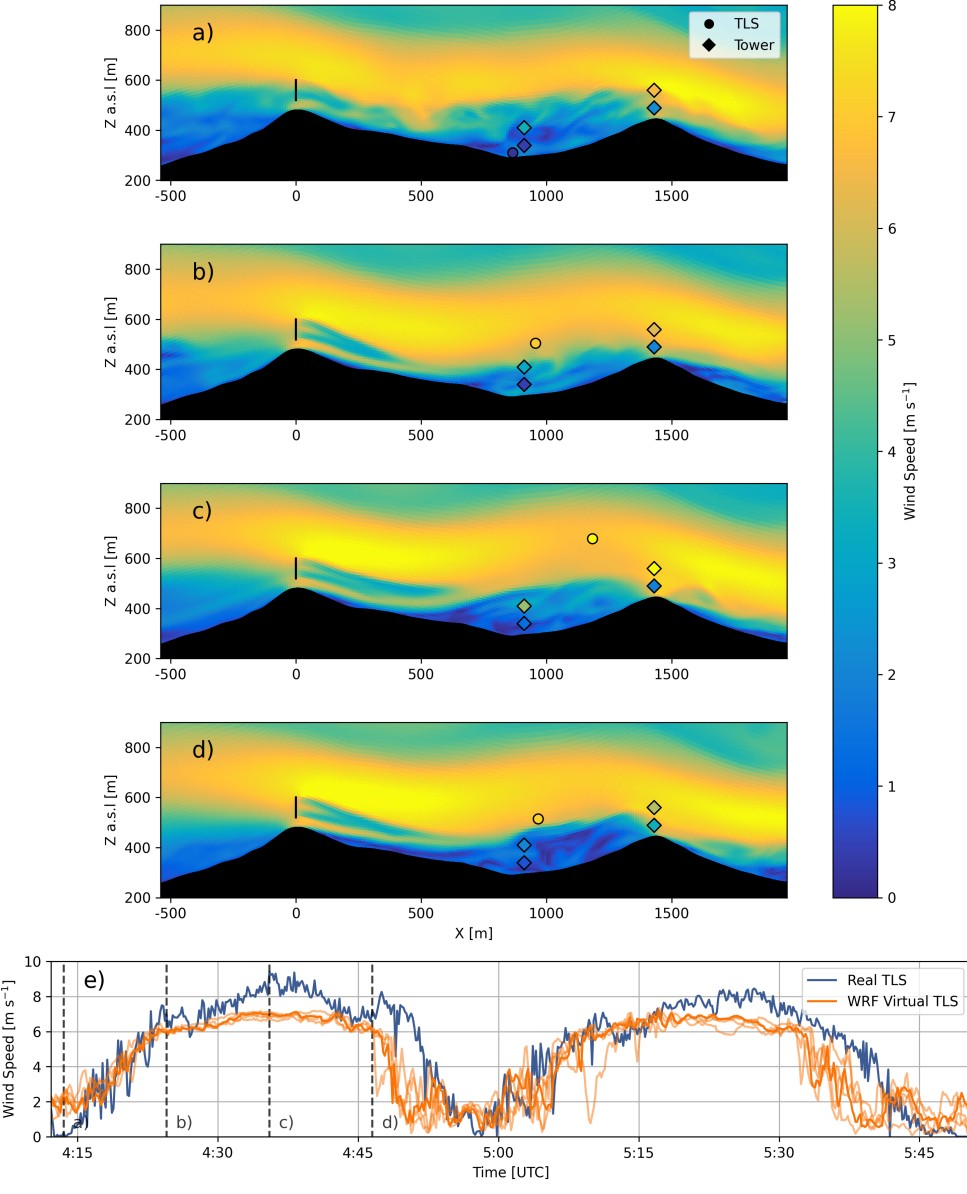

**Figure 7.** Transects of instantaneous wind speed for WRF with measurements from V_TSE09, NE_TSE13, and the TLS overlaid for (a) 04:13:30 UTC, (b) 04:24:30 UTC, (c) 04:35:30 UTC, and (d) 04:46:30 UTC and (e) comparison of wind speed between the TLS data and the virtual TLS in WRF-LES-GAD d05. The transects are aligned with the Wake Transect in Fig. 1. Virtual TLSs with an easting and northing position ±30 m are shown with a lighter shading and dashed lines indicate the times shown in the transects. An animation of model results for when the TLS is operational is included in the supplementary material (see Video 2, in the Video Supplement).





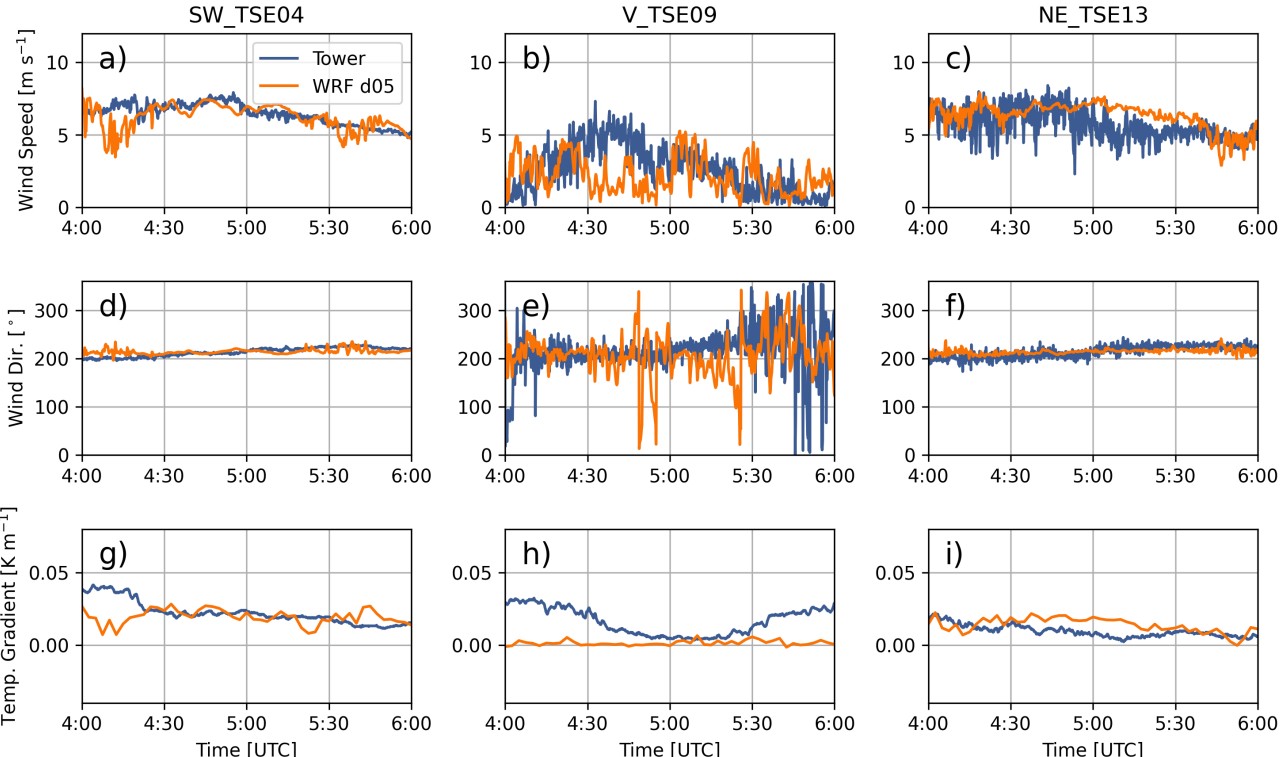

**Figure 8.** Comparison between meteorological tower data and WRF-LES-GAD d05 for 80 m wind speed (a-c), 80 m wind direction (d-f) and 100 m - 10 m temperature gradient (g-i) for the stable case study. SW_TSE04 and NE_TSE13 are located on ridges while V_TSE09 is located in the valley.

and 05:16 UTC and the innermost domain of the model, up to 5000 m above the surface. Errors in terms of bias and RMSE

between the soundings and the model are shown in Table 3.

For the first sounding, overall wind speed errors are small; however, as previously observed when comparing the model with measurements, the strength of the jet is underestimated. Using the velocity at the nose of the jet, we can calculate a Froude number for both the model and measurements at the beginning of the period of interest. The Brunt-Väisälä frequency can be estimated as $0.035$ s$^{-1}$ (see Fig. 3) and the mountain height is approximately 230 m. Using a free stream wind speed $U =$

$9.8$ m s$^{-1}$ for the sounding and $7.1$ m s$^{-1}$ for the model, the Froude numbers are 1.2 and 0.9, respectively. The lower Froude number for the model results in a shorter wavelength for the mountain wave, as was seen in Fig. 6. The model struggles to capture the near-surface wind direction in the valley, however this is primarily because wind direction is sensitive to fluctuations at lower wind speeds. The wind direction also deviates between 1-2 km above sea level (a.s.l) but the overall errors are small with very strong agreement in the upper atmosphere. There is a small negative bias for the potential temperature, but the height

of the stable layer is accurately captured by the model with an inversion located close to 600 m a.s.l. For the second sounding,



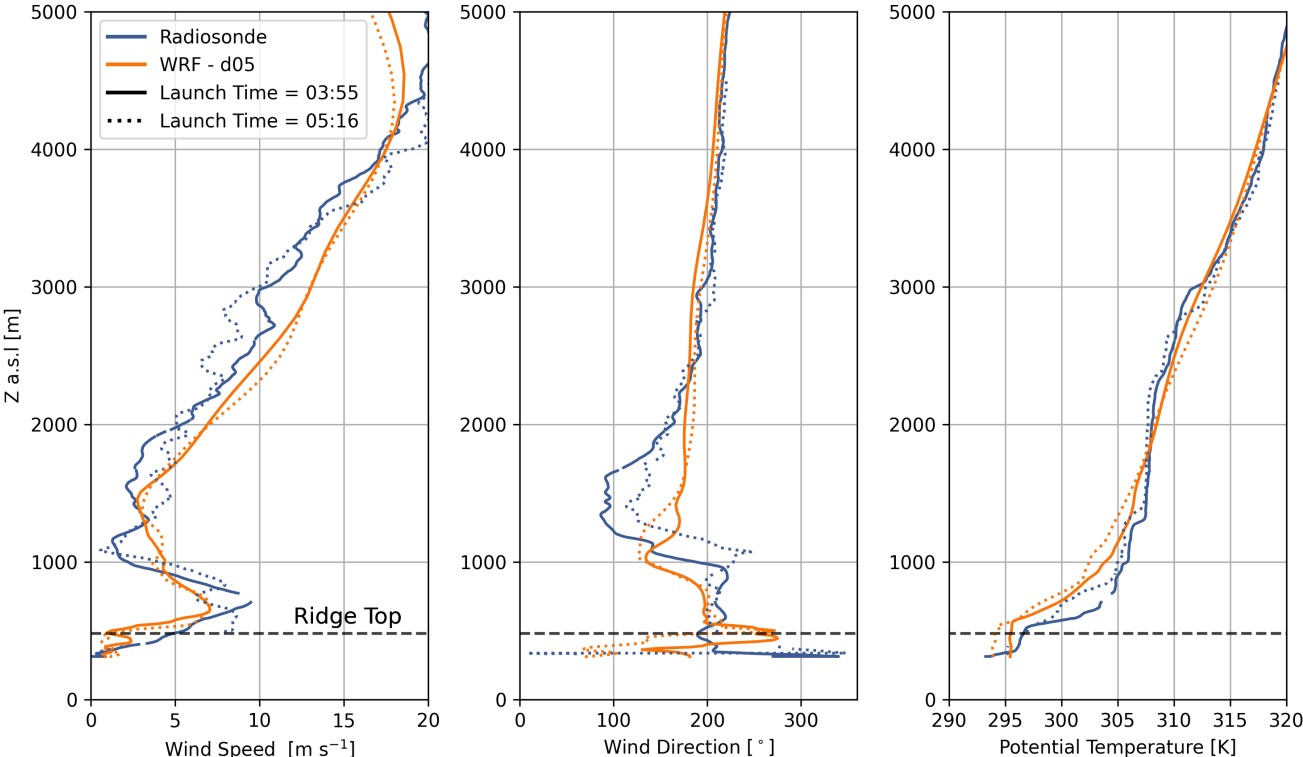

**Figure 9.** Comparison between the two sounding launched during the stable case study and WRF-LES-GAD d05 for wind speed, wind direction, and potential temperature.

overall wind speed errors are similarly small, although a small decrease in wind speed within the nose of the jet is observed in the sounding at about 800 m a.s.l. This decrease in wind speed could be the turbine wake or the striation of lower wind speed seen close to 600 m a.s.l in the multi-Doppler lidar scans (Fig. 6(b) and Fig. 6(d)). This decrease in wind speed does not emerge in the model. Similar to the first sounding, the wind direction deviates near the surface and between 1-2 km a.s.l

during the second sounding, but is largely captured aloft. The small negative bias for potential temperature is also apparent for the second sounding, but the height of temperature inversion is accurately captured close to 600 m a.s.l.

Time-averaged velocities highlight that the model predicts the overall characteristics of the flow and wind turbine wake behavior accurately when compared with lidar scans; however, there are minor discrepancies associated with the height and wavelength of the mountain wave as well as with the flow within the wave. Figure 10 shows 1 h (from 04:30 to 05:30 UTC)

time-averaged velocities for the stable case from the model and the DLR lidar scan along the Wake Transect (Fig. 1). The velocities from the model and DLR lidar have been projected along-transect as in Fig. 6. As mentioned in the previous section when looking at the soundings, the height of the mountain wave does not extend as high in the model compared to measurements. This is clear in Fig. 10, where the height of the wave ends close to 1200 m when there is flow in the opposite direction.





**Table 3.** Wind speed, wind direction, and potential temperature biases and errors between WRF-LES-GAD d05 and soundings for the stable case study.

|  | Sounding 1 | | Sounding 2 | |
| Parameter | Bias | RMSE | Bias | RMSE |
| --- | --- | --- | --- | --- |
| Wind Speed (m s$^{-1}$) | -0.52 | 1.58 | 0.77 | 2.34 |
| Wind Direction (°) | 2.2 | 32.9 | -4.6 | 36.6 |
| Potential Temperature (K) | -0.35 | 1.11 | -0.29 | 1.27 |

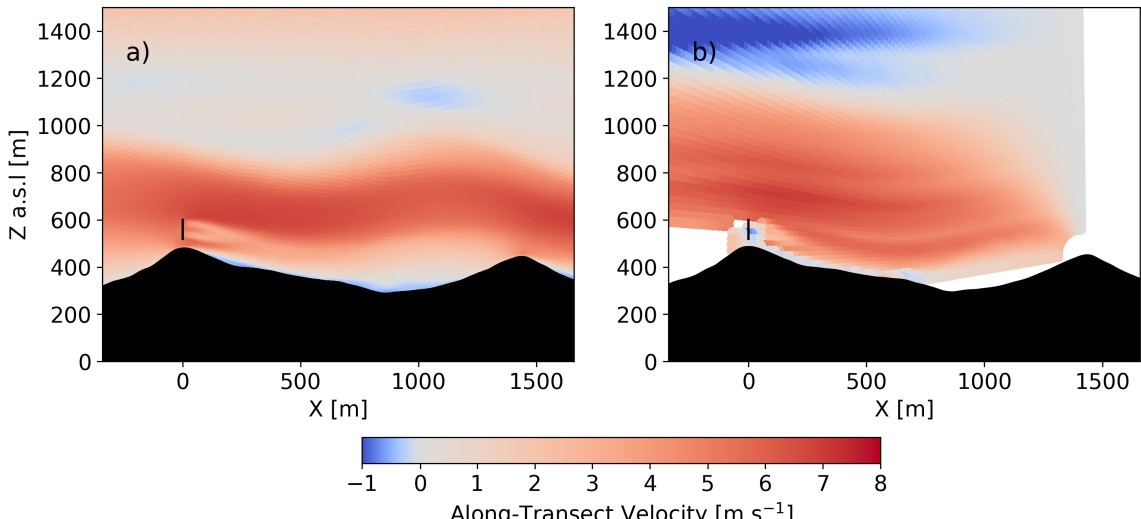

**Figure 10.** Transect of 1 h time-averaged along-transect velocity for the model (a) and DLR lidar (b) during the stable case study.

For the model, the height of the wave ends closer to 900 m. The lidar data also show striations of slower and faster wind speeds

within the wave, but these striations are not captured by the model. For both the model and the lidar scans, the wake propagates downward following the terrain into the valley. The velocity deficit dissipates more quickly in the observations compared to the model. The model's resolvable turbulent length scale is limited by the grid resolution, and for stable boundary layers this length scale can be small. It is possible that turbulence dissipation for the model is under predicted and this could be due to the different turbine model or due to errors from a number of other parameterizations.



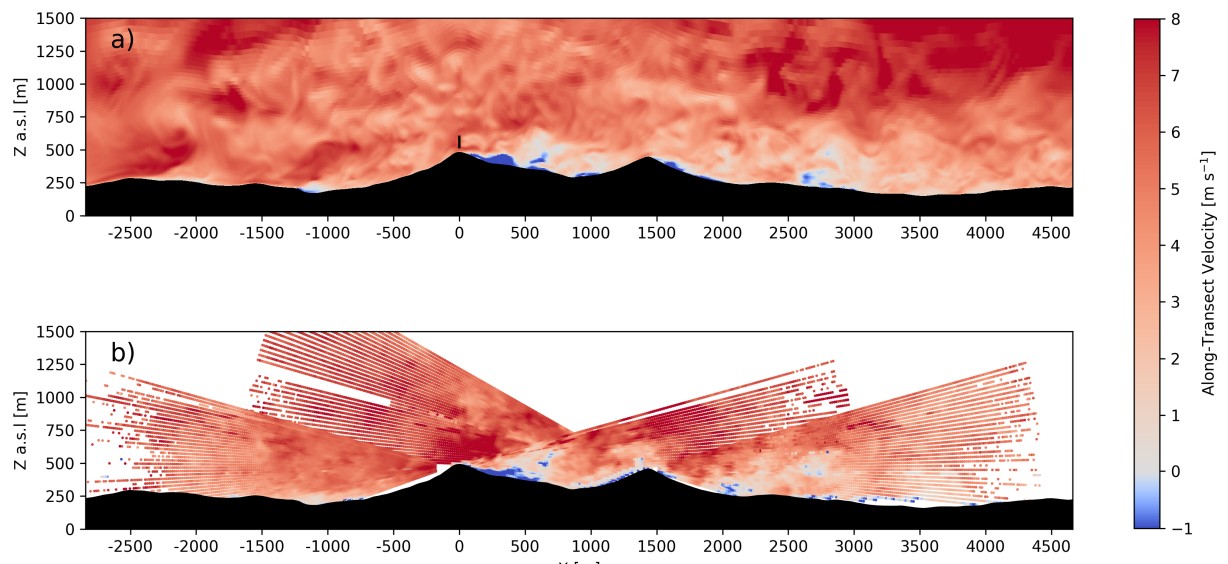

**Figure 11.** Instantaneous along-transect horizontal velocity during the convective case study for the model (a) and DTU multi-Doppler lidar scan (b) at approximately 13:20 UTC. The output from the model is instantaneous while the multi-Doppler measurements are over the course of a single scan, which takes about 24 s. The model transect is aligned with the Wake Transect and the lidar transect is aligned with the Offset Transect in Fig. 1. An animation of model results during the entire one-hour period are included in the supplementary material (see Video 3, in the Video Supplement).

## 4.2 Convective Case Study

The convective case on 13 May 2017, 13:00 - 14:00 UTC is much more turbulent compared to the stable case due to surface heating. Increased turbulent mixing and unstable stratification lead to a turbulent mountain wake in the lee of the first ridge, which forms a recirculation zone. WRF-LES-GAD and the multi-Doppler lidar scans can be similarly compared for the convective case as they were for the stable case (Fig. 6). Reverse flow near the surface in the lee of the first ridge is both predicted by the model and observed by the multi-Doppler lidar scans as seen in Fig. 11. Turbulent eddies are clearly visible over the entirety of the transects in Fig. 11(a) and Fig. 11(b).

The flow inside the valley and near the two ridges is highly dynamic but very accurately predicted by WRF-LES-GAD. Figure 12 shows the instantaneous wind speed from the model overlaid with measurements from V_TSE09 and NE_TSE13 at four different instances. A similar procedure as for Fig. 7 is followed here for Fig. 12. Note that the TLS was not operational during the convective case study and is therefore not available for comparison. Wind speeds are very slow in the lee of the first ridge, representing a mountain wake. Above this mountain wake, there is a strong shear layer with higher wind speeds in which the wind turbine wake propagates. While the flow is both dynamic and highly turbulent, measurements from V_TSE09 show



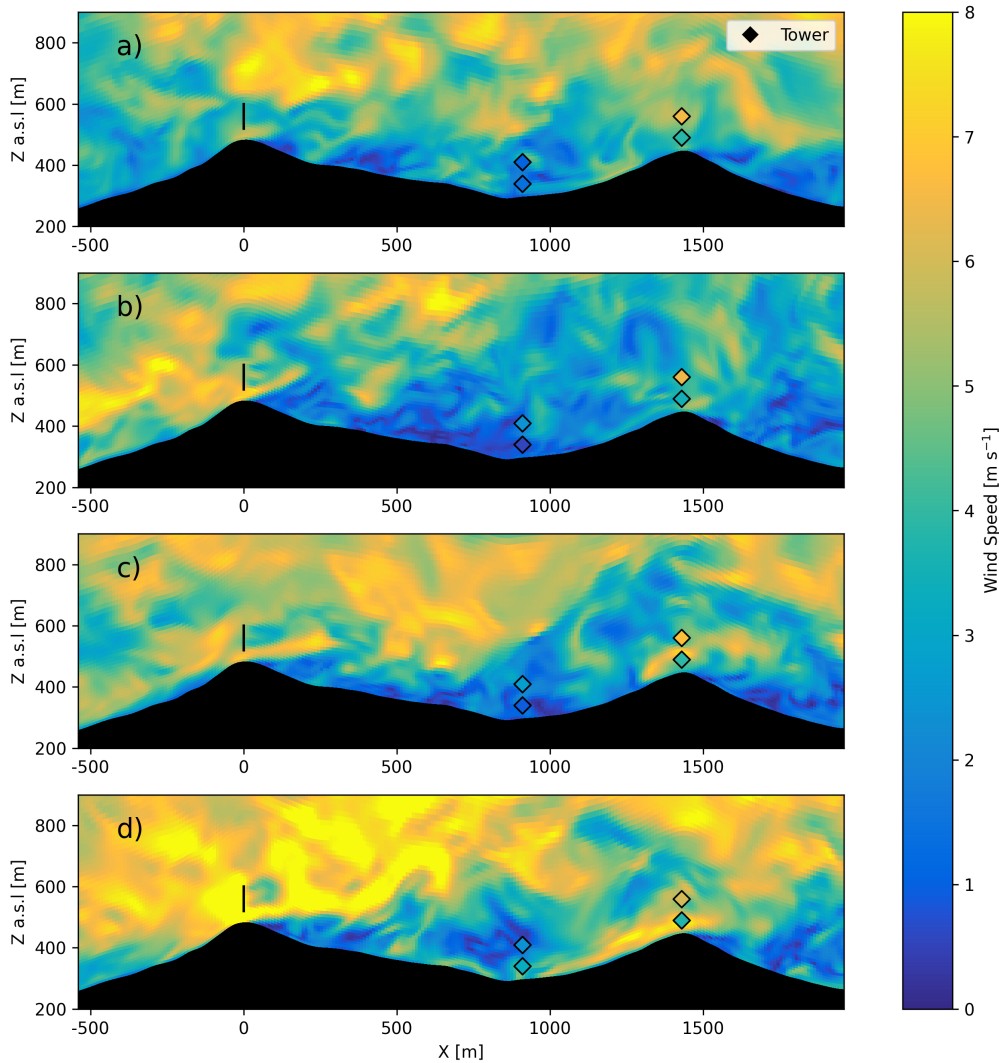

**Figure 12.** Transects of instantaneous wind speed for WRF with measurements from V_TSE09 and NE_TSE13 for (a) 13:13:30 UTC, (b) 13:24:30 UTC, (c) 13:35:30 UTC, and (d) 13:46:30 UTC. The transects are aligned with the Wake Transect in Fig. 1. An animation of model results during the entire one-hour period are included in the supplementary material (see Video 4, in the Video Supplement).

slow wind speeds both near the surface and at 80 m during all four instances. This indicates that V_TSE09 is generally within the mountain wake of the first ridge. Wind speeds at 80 m for NE_TSE13 are higher than wind speeds near the surface at all

four time instances, which the model generally, but not always agrees with.

The wind speed and wind direction are well-captured by the model on the ridges but with larger errors in the valley. Additionally, there is less variability in the potential temperature gradient for the model compared with observations, but biases are small. Figure 13 shows the 80 m wind speed and direction, and the 100 m - 10 m potential temperature gradient for the model



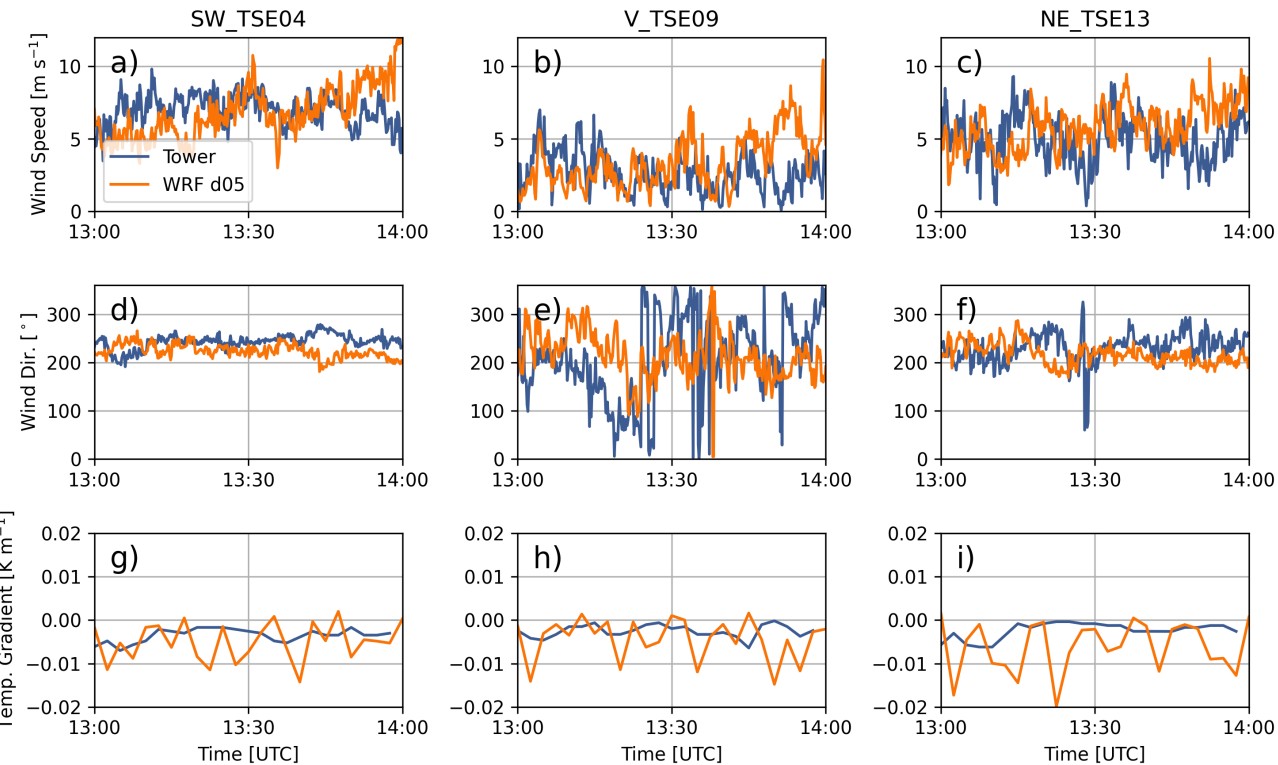

**Figure 13.** Comparison between meteorological tower data and WRF-LES-GAD d05 for 80 m wind speed (a-c), 80 m wind direction (d-f) and 100 m - 10 m temperature gradient (g-i) for the convective case study. SW_TSE04 and NE_TSE13 are located on ridges while V_TSE09 is located in the valley.

as compared with the towers on the southwest ridge, in the valley, and on the northeast ridge. During this time period, all three
metrics are relatively constant with the greatest variability in the wind speed and wind direction in the valley. Differences in
80 m mean wind speeds between the model and V_TSE09 are less than 1 m s$^{-1}$ and less than 1.5 m s$^{-1}$ for the other two
towers. Differences in 80 m mean wind direction are less than 15° on the ridges and less than 3° in the valley. Overall, the fluc-
tuations in measurements are much more pronounced in this convective case compared to the stable case (Fig. 8). Variability
in the inflow is slightly over-predicted with a standard deviation of 1.74 m s$^{-1}$ for the model and 1.16 m s$^{-1}$ for SW_TSE04.
365   Errors in terms of bias and RMSE between the met tower measurements at both 80 m and 10 m and the model are shown
in Table 4. During the period of interest, wind speed errors are on the order of 2 m s$^{-1}$ or less. The wind speed and direction
errors are, in general, larger than those for the stable case. The larger errors in the near-surface wind direction for the valley are
similarly due to the lower wind speeds with increased turbulence compared to the flow on the ridges. The temperature gradient
is much more variable in the model compared to the met towers, although the biases in the temperature gradient are small,
370   indicating that the average strength of thermal stratification is captured well throughout the period of interest.



**Table 4.** Wind speed, wind direction, and temperature gradient biases and errors between WRF-LES-GAD and meteorological tower measurements.

| Parameter | SW_TSE04 | | V_TSE09 | | NE_TSE13 | |
|---|---|---|---|---|---|---|
| | Bias | RMSE | Bias | RMSE | Bias | RMSE |
| 80 m Wind Speed (m s$^{-1}$) | 1.70 | 2.17 | 0.62 | 2.38 | -0.74 | 2.04 |
| 10 m Wind Speed (m s$^{-1}$) | -0.12 | 2.25 | 1.28 | 2.12 | 0.35 | 1.67 |
| 80 m Wind Direction (°) | -19.5 | 31.6 | 6.4 | 101.2 | -14.5 | 44.0 |
| 10 m Wind Direction (°) | -15.5 | 23.9 | 77.1 | 141.2 | -13.3 | 39.2 |
| Temperature Gradient (K m$^{-1}$) | -0.002 | 0.005 | -0.002 | 0.005 | -0.004 | 0.007 |

**Table 5.** Wind speed, wind direction, and potential temperature biases and errors between WRF-LES-GAD d03 and sounding for the convective case study.

| Parameter | WRF - d03 VL1 | | WRF - d03 VL2 | |
|---|---|---|---|---|
| | Bias | RMSE | Bias | RMSE |
| Wind Speed (m s$^{-1}$) | 0.80 | 1.66 | 0.87 | 1.87 |
| Wind Direction (°) | 1.17 | 10.42 | -6.99 | 14.78 |
| Potential Temperature (K) | -0.24 | 1.07 | -0.17 | 0.51 |

While no sounding was launched during the convective period of interest, a sounding was launched on 13 May 2017 at 11:15 UTC. Domains d04 and d05 are not launched until after this time, but d03 has been running for over eight hours at this point. Figure 14 compares the wind speed, wind direction, and potential temperature profiles from WRF-LES-GAD d03 and those obtained by the sounding. In Fig. 14, there are two soundings for the model, one sounding virtually launched at the same time as the real sounding (11:15 UTC) and another sounding virtually launched an hour later (12:15 UTC). The potential temperature profile for the first sounding in the model is under predicted by close to 2 K in the lower atmosphere; however, the potential temperature profile in the second sounding from the model is well-mixed and matches the observed temperature profile much better. The RMSE for potential temperature is half of that for the second sounding compared to that of the first sounding (see Table 5). This indicates that there is a short delay in the model when capturing the thermal evolution of the atmosphere. The wind speed and wind direction profiles show strong agreement for both soundings, especially in the upper atmosphere. The wind speed biases are less than 1 m s$^{-1}$; however, the RMSEs are closer to 2 m s$^{-1}$ indicating less variability in the model. This lack of variability is not unexpected for d03 because the grid resolution is only 150 m.

The time-averaged velocities demonstrate that the recirculation zone in the lee of the first ridge is very well predicted in terms of size, and that the wind turbine wake, in general, deflects above this recirculation zone. We can similarly compare the model with the DLR lidar, as was done for the stable case (Fig. 10), to characterize the wind turbine wake interaction with the recirculation zone. This comparison is shown in Fig. 15 over a 1 h time average (from 13:00 to 14:00 UTC). We

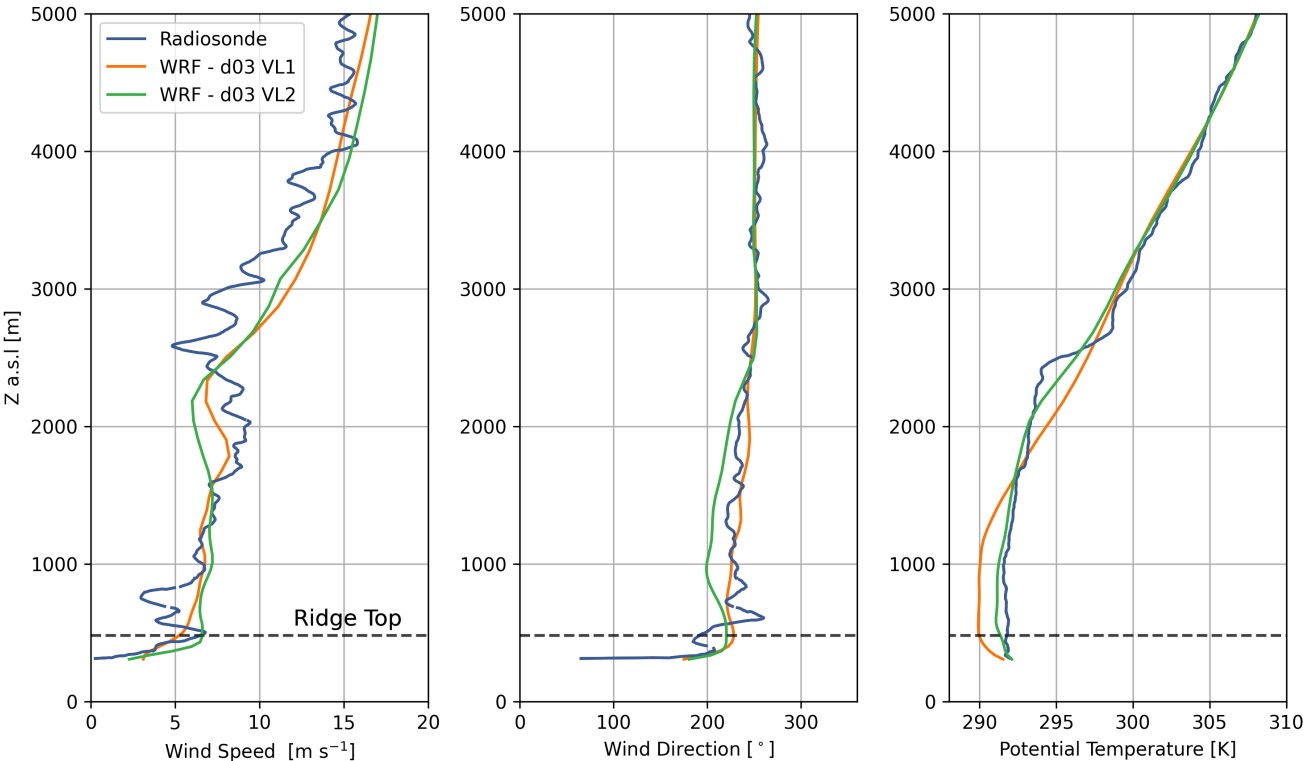

**Figure 14.** Comparison between the sounding, launched at 11:15 UTC, and soundings virtually launched (VL) in WRF-LES-GAD d03 for wind speed, wind direction, and potential temperature. WRF - d03 VL1 corresponds to a virtually launched sounding at 11:15 UTC and WRF - d03 VL2 corresponds to a virtually launched sounding at 12:15 UTC.

define recirculation as flow (along-transect horizontal velocity) in the upwind or negative direction. From Fig. 15, we can see the recirculation zone in the lee of the first ridge, which extends nearly 500 m into the valley for both the model and the measurements by the lidar. Additionally, there is also an area of slower wind speeds above the recirculation zone that can be classified as a mountain or ridge wake. For both the model and measurements, the wake does not mix with the recirculation zone or mountain wake and deflects upwards.

## 4.3 Wind Turbine Wake Behavior

For the complex Perdigão terrain, Menke et al. (2018) and Wildmann et al. (2019) showed that wind turbine wakes in stable stratification can be observed up to 10 rotor diameters (D) downstream, following the terrain into the valley. Our model agrees with these observations. Figure 16 shows instantaneous wind speed at approximately 80 m above the terrain for the stable case (14 June 2017 04:37:30 UTC) and the convective case (13 May 2017 13:33:20 UTC). We can see the wake signature in terms of a wind speed deficit persisting over 700 m into the valley for the stable case. The wake for the convective case dissipates

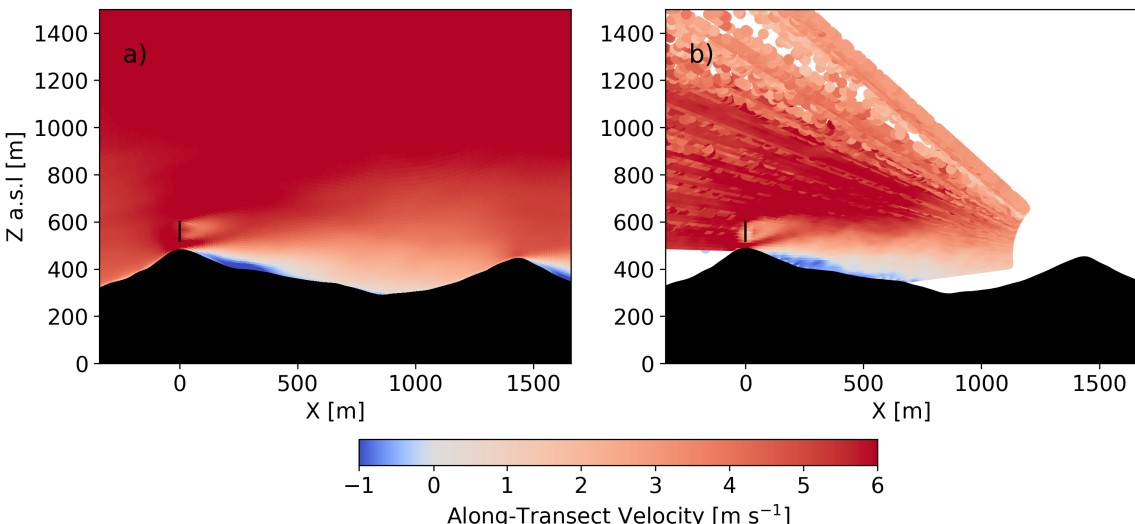

**Figure 15.** Transect of 1 h time-averaged along-transect velocity for the model (a) and DLR lidar (b) during the convective case study. The transects are aligned with the Wake Transect in Fig. 1.

much more quickly; however, this terrain-following plan-view makes it difficult to visualize the wake's extent because the wake deflects upwards in the convective case.

Three-dimensional visualizations provide insight into the wind turbine wake advection and direction downstream as the flow evolves over the first ridge and through the valley. Figure 17(a) and Fig. 17(b) show a volume rendering of wind speed at 04:37:30 UTC for the stable case. Note that the colorbar has been designed to highlight the velocity deficit imparted by the wind turbine rotor. We can see in Fig. 17(a) that the coherent, tubular structure that is the wind turbine wake propagates downward into the valley following the terrain. At wind speeds highlighted by the volume rendering, the wind turbine wake is

the dominant structure compared to any background turbulence in the flow field. In Fig. 17(a), the wake persists well into the valley and only begins to dissipate as it reaches the second ridge. Figure 17(b) shows how the wake meanders as it propagates into the valley. Wind veer causes the wake to deflect to the north near the second ridge.

    Turbulent structures dominate the flow field in the convective case. A volume rendering of wind speed at 13:33:20 UTC for the convective case is shown in Fig. 17(c) and Fig. 17(d). A similar design of the colorbar that was done for Fig. 17(a) and

Fig. 17(b) is done here for Fig. 17(c) and Fig. 17(d) except that the visible range of wind speeds is shifted from 3.0-5.0 m s$^{-1}$ to 3.5-5.5 m s$^{-1}$. Additionally, data to the south and west of the wind turbine have been omitted for clarity. The wake structure is not nearly as coherent and dissipates much more quickly in the convective case compared to the stable case. Figure 17(c) clearly shows the wake as the dominant feature on top of the first ridge. Directly behind the rotor, the wake is level but is then



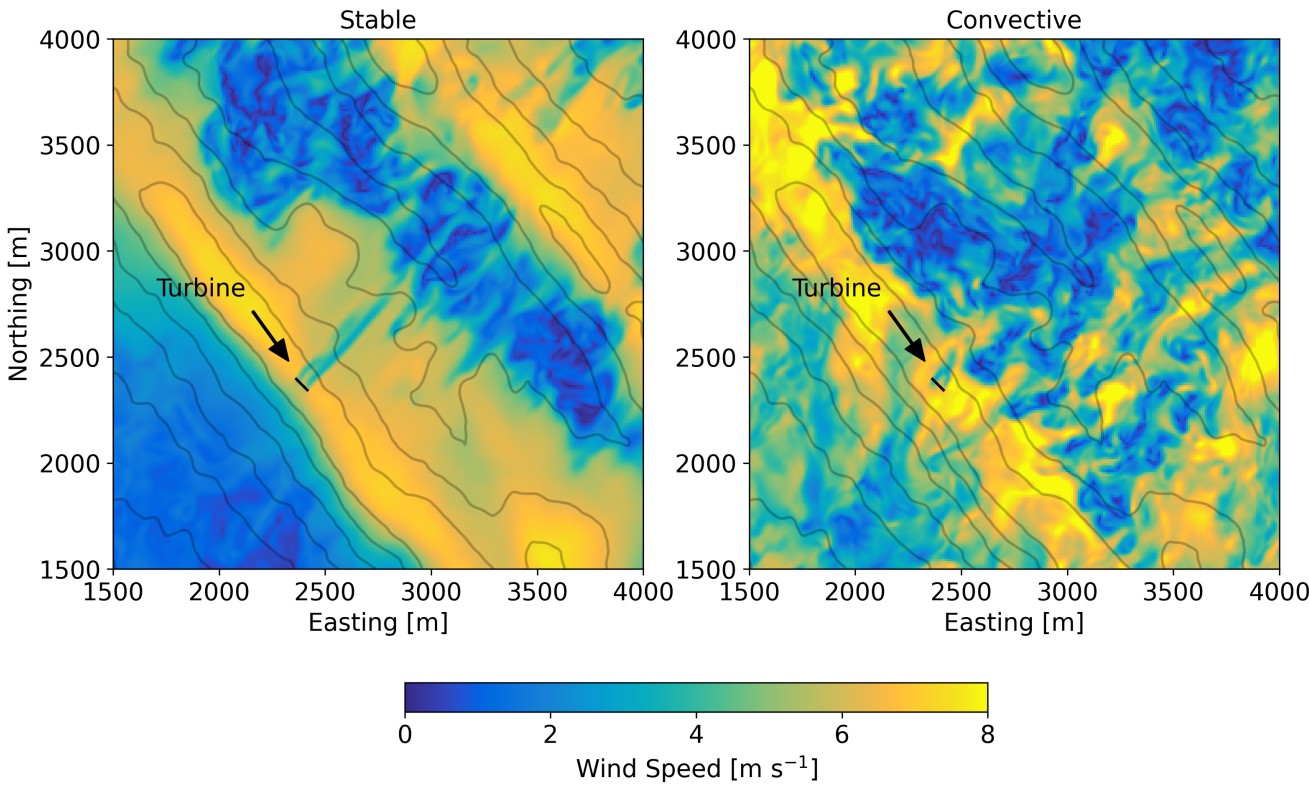

**Figure 16.** Instantaneous wind speed at 80 m above the terrain for the stable case study at 14 June 2017 04:37:30 UTC and for the convective case study at 13 May 2017 13:33:20 UTC. Contours represent 50 m changes in elevation.

deflected upwards. Background turbulence begins to dominate in the lee of the first ridge and into the valley. The camera angle
used in Fig. 17(d) makes it so the wake is difficult to distinguish from any background turbulence except very close to the rotor.

The recirculation zone for the convective case and the mountain wave for the stable case modulate the flow behavior in the valley. The evolution of these phenomena, their interaction with the wind turbine wake, and any spatial heterogeneity in the flow can be visualized using $y - z$ cross sections at different downstream distances. Figure 18 shows 1 h time-averaged wind speed for the two stability cases at distances of 1D, 2D, 3D, and 4D along the Wake Transect (see Fig. 1). For the convective
case, we can see that as the flow follows the terrain down into the valley, the recirculation zone and mountain wake rise in height. The wake structure is circular and much more coherent at distances of 1D and 2D compared to further downstream; however, a wind speed deficit is still clearly visible above the recirculation zone further downstream. The wake also drifts slightly in the positive Y-direction, but this is due to the slight misalignment between the Wake Transect and the mean wind direction. For the stable case, the wake propagates downward, following the terrain, above the slower near-surface winds.
Additionally, we see the upper half of the wake veer in the positive Y-direction leading to an ellipsoid shape for the wake. This stretching of the wake due to the veer in the inflow is characteristic of stable conditions (Lundquist et al., 2015; Abkar et al.,

**Figure 17.** Three-dimensional volume render of wind speed for the stable case at 04:37:30 UTC (a and b) and for the convective case at 13:33:20 UTC (c and d). For the stable case, only wind speeds between 3.0 and 5.0 m s$^{-1}$ are shown and the wind speeds at the first five vertical levels have been removed for clarity. For the convective case, only wind speeds between 3.5 and 5.5 m s$^{-1}$ are shown and data to the south and west of the turbine have been removed for clarity. The view point in (a) and (c) is from the southeast looking down the valley to the northwest. The view point in (b) and (d) is southwest of the wind turbine looking to the northeast. Visualization made using VAPOR (Clyne et al., 2007).





2016; Bromm et al., 2017; Churchfield and Sirnivas; Englberger and Dörnbrack, 2018; Bodini et al., 2017), while the amount of veer in the wake depends on the shape and magnitude of the inflow veer (Englberger and Lundquist, 2020).

To further quantify wake behavior in the Perdigão terrain, we determine the wake center locations downstream of the wind turbine. Using the model data, we extract 1 h time-averaged velocity profiles at distances of -2D, -1D, 0D, 1D, 2D, 3D, and 4D from the rotor along the Wake Transect. Since the terrain is similar on parallel transects, we also extract 1 h time-averaged velocity profiles at a lateral distance of 150 m parallel to the Wake Transect at the same locations upstream and downstream of the turbine. These velocity profiles 150 m further away do not capture the wake velocity deficit while the profiles directly behind the rotor do. The velocity deficit can then be calculated by taking the difference between the profiles 150 m further away

and the profiles that include the turbine wake. This difference results in a positive velocity deficit for the wake. The vertical wake centers can then be determined by finding the vertical location at which the velocity deficit profile is greatest.

The velocity deficit profiles show that the wake tracks upwards in the convective case and follows the terrain for the stable case (Fig. 19). The wake deflects upwards above the hub-height by 11 m 1D downstream and up to as high as 54 m for 4D downstream for the convective case. For the stable case, the wake deflects downwards nearly 100 m at 4D downstream. An

acceleration occurs in the velocity profiles for both case studies below the wake. This behavior was seen in the LES simulations of Vanderwende et al. (2016). This speedup is more apparent for the stable case, but is still clear for the convective case in the near wake. While the speedup could be due to the small differences in terrain between the two transects, it is more likely due to flow being channeled beneath the wake and above the slower near-surface wind speeds; this is consistent with the observation that this near-surface speedup occurred in LES of wakes in flat terrain (Vanderwende et al., 2016). The differences in the

velocity deficit upstream of the wind turbine are due to induction/blockage (Forsting and Troldborg, 2015; Bleeg et al., 2018; Segalini and Dahlberg, 2020) or from terrain effects.

## 5 Conclusions

The WRF multi-scale modeling framework has been used to successfully simulate wind turbine wake propagation in complex terrain under different atmospheric stability conditions. WRF has been augmented to include a generalized actuator disk (GAD)

model to represent individual wind turbine rotors. We used two configurations of WRF-LES-GAD. The first configuration is a single-domain, semi-idealized large-eddy simulation setup in which wind speed, wind direction, and potential temperature profiles are specified and atmospheric stability is controlled by specifying a surface heat flux. The second configuration is a multi-scale, nested approach in which WRF-LES-GAD has five domains with grid resolutions of 6750, 2250, 150, 50, and 10 m, with the coarsest two grids being meso-scale simulations and the finest three as micro-scale LES. Note that the first

configuration was only used as a preliminary tool to determine whether the more computationally expensive multi-scale setup would likely capture the terrain-induced flow phenomena of interest.

This is, to the authors' knowledge, the first study in which the effects of highly complex terrain are combined with a wind turbine parameterization for real large-eddy simulations in non-neutral stability conditions to capture the complex interplay between terrain, atmospheric stability, and a wind turbine wake. This novel framework was applied to two case studies: a stable

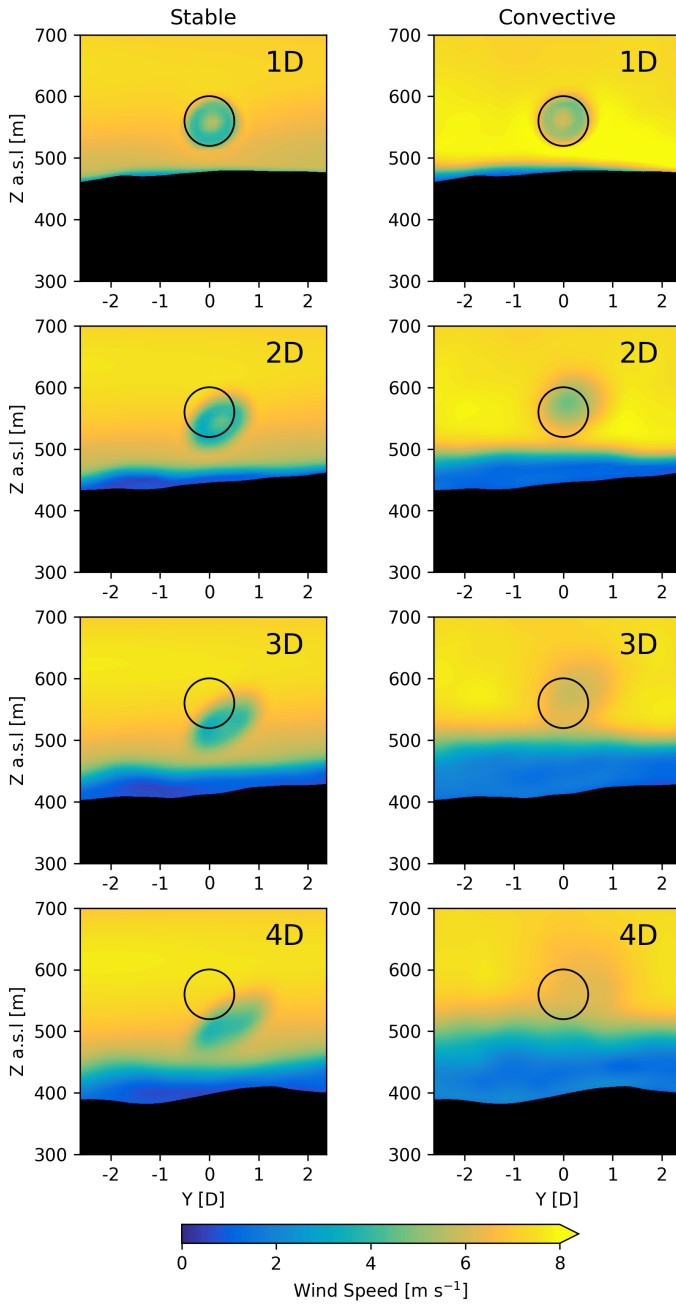

**Figure 18.** 1 h time-averaged wind speed at different downstream distances for the stable and convective case studies taken along the Wake Transect in Fig. 1. The black circle represents the circumference of the wind turbine rotor at a constant height a.s.l.

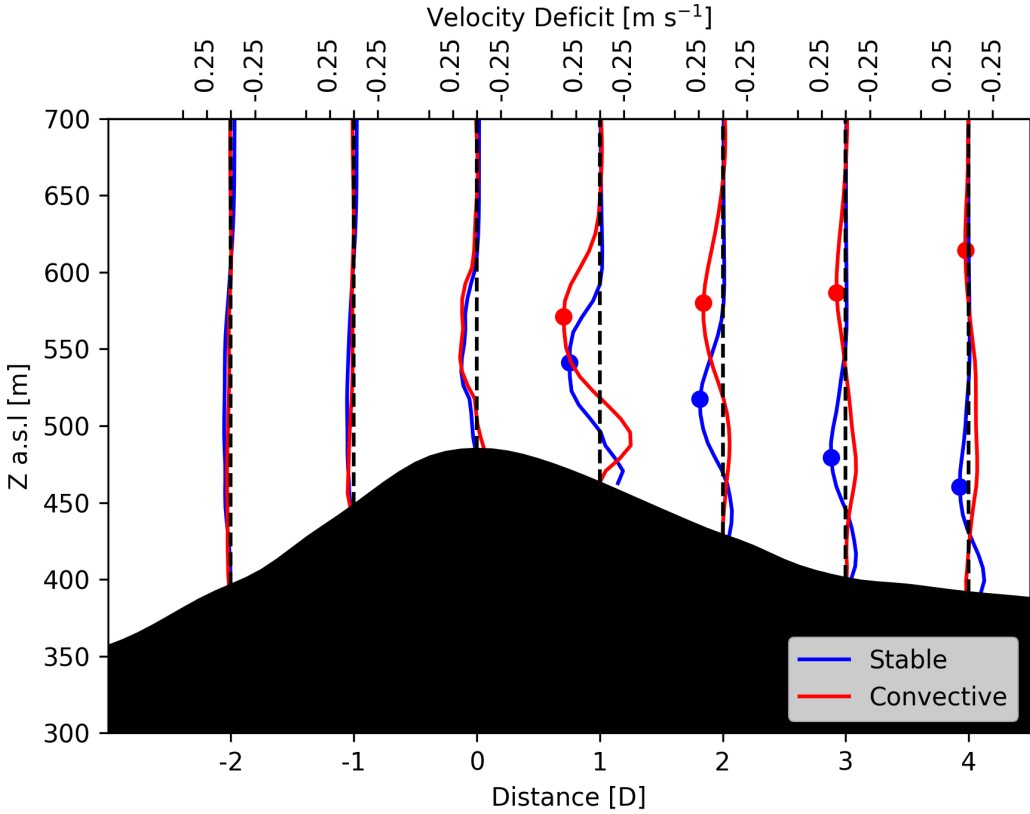

**Figure 19.** 1 h time-averaged velocity deficit profiles at different downstream distances for the stable and convective case studies.

case study where a mountain wave was present and a convective case study where a recirculation zone formed and interacted with the wake. Further, we validated our simulations by comparison to field observations with in situ and remote sensing instrumentation.

In the stable case, the wind turbine wake is advected downward following the terrain along with the mountain wave. The general characteristics of the mountain wave are well-captured by the model over the extent of the roughly 8 km domain in comparisons with meteorological towers, a tethered lifting system, radiosonde, and lidar data. The wind speeds within the wave were slightly underestimated which resulted in under-prediction of the wavelength for the mountain wave. Notably, the mountain wave caused the wake to deflect downwards into the valley. At four rotor diameters downstream of the wind turbine, the wake deflected downwards on average by nearly 100 m below the hub-height. Further, the stable wake stretches from a circular wake into an ellipse due to veer in the stable wind profile. Downwind from the turbine, a near-surface jet is created between the surface and the wake with winds increasing by ~0.25 m s$^{-1}$.

In the convective case, the wake is lofted above the terrain and above the original elevation of the wake as a result of the recirculation zone that forms in the lee of the first ridge. During this period, the near surface flow can be characterized by



a negative potential temperature gradient indicating convective or unstable conditions. The TLS was not operational during this period; therefore, the model was strictly validated against the meteorological towers and a single sounding, followed

by comparisons with lidar scans. With unstable thermal stratification, a mountain wake formed in the lee of the first ridge. Averaged over the hour, we observed a 500 m long recirculation zone in the valley. The formation of a recirculation zone resulted in the wind turbine wake deflecting upwards. The lack of mixing between the wake and recirculation zone resulted in the wake deflecting an average of 54 m above hub-height at four rotor diameters downstream. In the convective case, the wake maintains a circular structure downwind but diffuses more rapidly than the stable case due to increased ambient turbulence.

Small speedups of ~0.1 m s$^{-1}$ also occur under the wake as it propagates downwind.

The WRF-LES-GAD framework presented here exhibits remarkably good agreement with field observations during the Perdigão field campaign, making it well-suited to examine the role of complex flow phenomena on wind turbine and wind farm dynamics. The ability of the model to capture different wind turbine wake behavior over complex terrain under stable and convective conditions indicates the model's ability to integrate both mesoscale and micro-scale flow phenomena which

influence the wake behavior. Further studies of interest at the Perdigão site may include comparisons with observed turbulence quantities. Of particular interest is the effect of turbulence closure models (Kirkil et al., 2012; Zhou and Chow, 2012) on the detailed behavior of the wind turbine wake and background turbulence. Additionally, it is important to examine the role of turbulence closure models on wake dissipation and recovery. The present study demonstrates the viability of WRF-LES-GAD to model not only flow in complex terrain, but also the flow's interaction with wind turbine wakes. As more turbines are sited in

complex terrain, WRF-LES-GAD will be a useful tool for modeling wake dynamics in such environments, and under different atmospheric forcing regimes.

*Data availability.* The full WRF-LES-GAD simulation data is on the order of terabytes but subsets of the data can be shared upon request. The input files are available online at https://doi.org/10.5281/zenodo.4711606

*Video supplement.* The following is available online at https://doi.org/10.5281/zenodo.4711606, Video 1: Multi-scale simulation of a wind

turbine wake modulated by a mountain wave in stable atmospheric conditions, along-transect velocity for the entire domain. Video 2: Multi-scale simulation of a wind turbine wake modulated by a mountain wave in stable atmospheric conditions, wind speed and zoomed into the valley. Video 3: Multi-scale simulation of a wind turbine wake modulated by a recirculation zone in convective atmospheric conditions, along-transect velocity for the entire domain. Video 4: Multi-scale simulation of a wind turbine wake modulated by a recirculation zone in convective atmospheric conditions, wind speed and zoomed into the valley.

*Author contributions.* Writing—original draft preparation and visualization, ASW; writing—review and editing, JMTN, RSA, JDM, JKL, and FKC; methodology, software, validation, and formal analysis, ASW, JMTN, RSA, and JDM; conceptualization, ASW, JMTN, and FKC;





investigation and data curation, JKL. All authors contributed with critical feedback on this research and have read and agreed to the published version of the manuscript.

*Competing interests.* The authors declare that they have no conflict of interest.

*Acknowledgements.* Correspondence with Norman Wildmann from DLR and Robert Menke of DTU regarding their lidar datasets was greatly appreciated. This material is based upon work by ASW supported by the National Science Foundation Graduate Research Fellowship Program under Grant No. DGE 1752814. The authors are grateful for support from National Science Foundation Grants AGS-1565483 and AGS-1565498. The Perdigão was primarily funded by the U.S. National Science Foundation, European Commission's ERANET+, Danish Energy Agency, German Federal Ministry of Economy and Energy, Portugal Foundation for Science and Technology, U.S. Army Research

Laboratory, and the Israel Binational Science Foundation. Perdigão would not have been possible without the alliance of many personnel and entities, which are listed in the supplemental material of Fernando et al. (2019). The TLS data presented here were collected by Ludovic Bariteau, Dr. Jessica Tomaszewski, Dr. Nicola Bodini, and Patrick Murphy. RSA's and JDM's contributions were prepared by Lawrence Livermore National Laboratory under Contract DE-AC52-07NA27344, with support from the U.S. DOE Office of Energy Efficiency and Renewable Energy Wind Energy Technologies Office. This work was authored [in part] by the National Renewable Energy Laboratory,

operated by Alliance for Sustainable Energy, LLC, for the U.S. Department of Energy (DOE) under Contract No. DE-AC36-08GO28308. Funding provided by the U.S. Department of Energy Office of Energy Efficiency and Renewable Energy Wind Energy Technologies Office. The views expressed in the article do not necessarily represent the views of the DOE or the U.S. Government. The U.S. Government retains and the publisher, by accepting the article for publication, acknowledges that the U.S. Government retains a nonexclusive, paid-up, irrevocable, worldwide license to publish or reproduce the published form of this work, or allow others to do so, for U.S. Government

purposes. High-performance computing support from Cheyenne (doi:10.5065/D6RX99HX) was provided by NCAR's Computational and Information Systems Laboratory, sponsored by the National Science Foundation.



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
