# Peer review of "Meso- to micro-scale modeling of atmospheric stability effects on wind turbine wake behavior in complex terrain"

_Wind Energy Science, 2021_

## Author Comment (AC2)

Please note that references to line numbers are for the revised manuscript.

**Scientific questions:**

1. Can you comment on how representative the specific case studies (stable in combination with mountain waves, unstable in combination with a recirculation zone) are for the wind conditions at Perdigão and for complex terrain in general? Do you think the conclusions in terms of wind-turbine wake behavior (terrain following in stable, deflecting upwards due to recirculation) will hold for all stable/unstable wind conditions at this site and for complex terrain in general?

Mountain waves occurred for almost 50% of the nights during the intensive observation period (Fernando et al. 2018). For recirculation, Menke et al. (2019) found that reverse flow with wind speeds greater than 0.5 m/s occurred over 50% of the time when the wind direction was perpendicular to the ridges. We have added these details into the manuscript at lines 130-132.

We expect that the conclusions in terms of the wind turbine wake behavior would hold for most convective and stable atmospheric conditions at the Perdigão site as long as the phenomena of interest (recirculation zones and mountain waves) are present. Other phenomena could be modeled using WRF-LES-GAD to examine wind turbine wake behavior in other cases. With regards to other sites, the wind turbine wake behavior would depend largely on the vegetation (surface roughness) and steepness of the terrain. If the previously mentioned parameters are similar to those in the examined case studies, similar conclusions for wake behavior in convective and stable conditions could be made. We have added this discussion to the manuscript in lines 514-518.

2. The inverse of the Froude number as defined in Equation 1 is often called the non-dimensional mountain height and it represents the ratio of the mountain height to the (vertical) wave length of the mountain wave. When discussing the case where the mountain wave length is shorter than the width of the mountain (line 133-134), wouldn't it be more appropriate to use a Froude number based on the width rather than the height of the mountain like for example in the book of Stull (1988, section 14.2.3 Flow over hills). Obviously, this also affects figure 3 and later calculations of the Froude number in section 4.1.

We thank the reviewer for bringing up this interesting point. Since we are dealing mostly with flow over the mountain rather than flow around the mountain, we believe the Froude number based on the mountain height is more appropriate. Additionally, the Froude number defined using the mountain width is for an isolated hill as opposed to an extended ridge as is the case for Perdigão.

The Froude number based on the ridge height has also been calculated for previous studies at Perdigão (Fernando et al. 2019, Palma et al. 2019).

The wavelength of a mountain wave is defined as lambda = 2*pi*U/N and is actually independent of both the mountain height and width. However, we can quickly calculate a Froude number based on width during the stable case. If the mountain width is 800 m, N is 0.035 /s, and U is 9.8 m/s, then Fr = 3.14*9.8/(0.035*800) = 1.1. Because of the proportion of the height and width of the ridge at Perdigão, calculations of the Froude number using either the mountain height or width are very similar.

The discussion comparing the wavelength to the mountain width has been removed to avoid any confusion. The sentence, lines 139-140, now reads: "For small Froude numbers (< 1), when wind speeds are low or the stability is very strong, the wavelength of the mountain wave is short resulting in weak mountain waves."

3. It is not entirely clear how you setup the semi-idealized simulations. What pressure gradient force do you impose, or alternatively what wind speed and direction do you enforce (and at what height)? Do you apply a negative surface heat flux in the stable case? What is the domain height of these simulations? Do you use any damping layers at the top? For how long did you simulate these idealized cases?

The semi-idealized simulations are set up using geostrophic forcing with a specified initial wind and potential temperature profile. The wind speed for the stable case comes from a sounding, while for the unstable case it is uniformly specified as 7 m/s. In both cases, the geostrophic wind direction is aligned with the x-axis, such that the forcing is entirely longitudinal. For the stable case, a cooling rate of -0.25 K/hr was used. However, the effect of this cooling rate is minimal as the simulation is only run for 10 minutes. The domain height is just above 1400 m and we use Rayleigh damping within 500 m of the domain top with a coefficient of 0.003 /s. These details have been incorporated into the semi-idealized modeling section in lines 198-209.

4. Line 182-183: What do you mean with the stratification is self-destructive? Do you mean the stable stratification turns into a constant temperature profile because of turbulent mixing? Does the simulation become unstable due to inertial oscillations? Something else?

Because the idealized simulation uses periodic boundary conditions, the flow will recirculate and the effects of the mountains and the GAD will induce mixing which will erode the stable

stratification. We have edited the manuscript to replace the term "self-destructive" with "eroded by turbulent mixing" in line 207.

5. Figure 12: Is there any averaging of model results or measurements? Under higher turbulent conditions, does it make sense to compare instantaneous velocity fields with the point measurements of the met towers given the chaotic nature of turbulence?

There is spatial averaging of 30 m in each spanwise direction (60 m total) in figure 12 but no time averaging. While we agree that exact agreement between the model and observations is not expected under highly turbulent conditions, we believe that this figure shows important qualitative agreement. The four transects included aim to be representative of the hour-long simulation period, highlighting that the general dynamics of the flow are captured despite differences due to the chaotic nature of turbulence.

6. Line 361 "Differences in 80m mean wind speeds … are less than 1 m/s …": Is this the difference averaged over the entire time horizon, or the root-mean square, or something else? When I look at figure 13b around 13:55, I obviously see differences of more than 1m/s. Similar for the discussion of the differences in the wind direction and also for the plots for stable conditions. You seem to be talking about a certain average difference being below a certain value, but the instantaneous differences are clearly higher than this value and it is not clear how the averaging is done. Please clarify.

The reviewer is correct in that instantaneous differences can be large and this is due to the previously noted chaotic nature of turbulence. This is why we have included both bias and RMSE as metrics comparing WRF-LES-GAD with observations. The RMSE captures the cumulative magnitude of the errors over the entire simulation period while bias captures the average behavior. To clarify the text, the discussion regarding mean metrics has been removed. In line 392, the variability quantified in terms of standard deviation now explicitly states that the standard deviation is calculated over the hour-long period. Additionally, errors displayed in Table 4 are discussed in the sentences that follow in the manuscript (lines 394-399).

7. Line 445: The blockage addressed in the papers by Meyer-Forsting, Bleeg and Segalini refers to the global blockage effect far upstream of large wind farms that arises due to the combination of many wind turbines. Here, you only have one turbine and you are only looking at

> 2D upstream, so I would say that turbine induction might be playing a role, not blockage. I don't think it is appropriate to refer to these wind-farm blockage studies here, so consider removing these references.

We have removed the blockage references and added two references that provide more detail on wind turbine induction zones.

Medici, D., Ivanell, S., Dahlberg, J.-A., and Alfredsson, P. H.: The upstream flow of a wind turbine: blockage effect, Wind Energy, 14, 691–697, https://doi.org/https://doi.org/10.1002/we.451, 2011.

Simley, E., Angelou, N., Mikkelsen, T., Sjöholm, M., Mann, J., and Pao, L. Y.: Characterization of wind velocities in the upstream
induction zone of a wind turbine using scanning continuous-
wave lidars, Journal of Renewable and Sustainable Energy, 8, 013 301, https://doi.org/10.1063/1.4940025, 2016.

**Minor questions and technical comments:**

> Line 148: Please mention at least once how UTC relates to local time at this particular site. Maybe it is even more interesting to mention what time corresponds to sunrise.

The local time has been added in sections 2.2.1 and 2.2.2 at lines 167 and 177, respectively. In section 2.2.1, the local time corresponding to sunrise has also been added in lines 167-168.

> Line 148-149: Please mention what value of the wind direction corresponds to a direction perpendicular to the ridge such that the statement can actually be assessed by the reader.

A southwesterly wind of 215 deg. corresponds to being perpendicular to the ridges. This has been added in line 169.

> Line 149 reference to Fig 2a and 2b: I don't think this reference is correct. Figures 2a and b show wind speed in stable and convective conditions, while the sentence was talking about the wind direction.

The reference should be just Fig. 2c which refers to the wind direction during the stable case. This has been fixed.

> Line 155-156 and figure 2d: Wouldn't it make more sense here to use an actual stability parameter like the Obukhov length L or z/L to assess the atmospheric stability? Especially when talking about moderately convective conditions, how can you assess that based solely on the potential temperature gradient?

Assessing atmospheric stability in complex terrain is an active research area. Using a gradient Richardson number is difficult because of the effect of terrain-induced flow-speedup over the ridge affecting the shear terms as mentioned by Menke et al. (2019). Additionally, Bodini et al. (2020) calculated the Obukhov length at Perdigão and found that it was not very powerful in predicting dissipation rate. However, we have calculated the Obukhov length and added it to Fig. 2 along with new discussion regarding its limited applicability for assessing atmospheric stability in complex terrain (lines 147-157). Additionally, we have removed "moderately" from this section since our primary intention is to say that the case is convective, rather than stable.

> Line 158 reference to Fig 2d: Again this reference seems to be a bit misplaced as it refers to the wind direction plot, not the wind speed plot.

The reference should be Fig 2b and this has been fixed.

> Fig2: Please mention hub height of 80m (?) and measurement points for the temperature gradient (10m and 100m?) in the caption for clarity.

This has been added in the caption of Fig. 2.

> End of line 188: surface roughness length should be in units of meters.

Units have been added.

> Line 291, 356, and many others : You often start a new paragraph without clearly indicating that you will be talking about a new figure. For instance on line 291, are you still referring to figure 7, or are you rather referring to figure 8 or table 2?

In lines 292, 318, 333, 357, 377, 387, and 412, we have added explicit references to figures to improve readability.

> Table 2 and 3: the results in these tables are not really discussed it seems. Can you somehow use these results when assessing/discussing the results of related figures?

For Table 2, we have added language talking about the RMSE of the temperature gradient in line 331. This is in addition to the specific numbers already mentioned and discussed in the paragraph from lines 324-332.

Specific numbers from Table 3 have been weaved into the paragraph from lines 338-356, where they can be related to figures.

> Line 312: How high above the actual surface is the nose of the jet which you use to calculate the Froude number? The Brunt-Vaisala frequency is based on the temperature gradient between 10m and 100m, but it is not clear whether the nose of the jet is at about the same height above the surface.

The nose of the jet is at 650 m a.s.l. in the model and 720 m a.s.l. in the measurements (which is 350 m and 420 m above the valley floor, respectively). This has been specified in the text at line 340 to avoid any confusion. The sentence has been rewritten as: "Using the velocity at the nose of the jet, which is at 650 m a.s.l. In the model and 720 m a.s.l in the observations, we can calculate a Froude number at the beginning of the period of interest."

> Line 320: How high is the inversion layer above the valley floor (600m asl but at what altitude is the valley floor)?

The valley floor is just under 300 m a.s.l. This has been added to the manuscript in lines 357-358:

"The small negative bias of -0.3 K for potential temperature is also apparent for the second sounding, but the height of temperature inversion is accurately captured close to 600 m a.s.l or 300 m above the valley floor."

Line 323: When talking about the striation of lower wind speed close to 600 asl in fig 6b and d, in what ranges of x can this be seen?

There is a coherent striation from the beginning of the transect until the rotor in the lee of the second ridge at x = 2500 m. This has been added to the manuscript in lines 351-352:

"This decrease in wind speed could be the turbine wake or the striation of lower wind speed seen close to 600 m a.s.l in the multi-Doppler lidar scans from the beginning of the transect until the rotor in the lee of the second ridge at X = 2500 m (Fig. 6(b) and Fig. 6(d))."

Line 357 "there is less variability in the potential temperature gradient for the model compared with observations": I'd say there is more variability in the model output compared with the observations (this is also mentioned later in the text). The statement seems to contradict the results shown in figure 13 g,h,i.

This should read "there is more variability…" which we have corrected in the manuscript at line 387.

Line 427: The year of the publication by Churchfield and Sirnivas is missing (also in the list of references).

This has been fixed.

We thank the reviewer for the comments. We think that the revised manuscript and the responses above help to address most of these concerns.

---

## Author Comment (AC3)

Please note that references to line numbers are for the revised manuscript.

Reviewer general comment: The current manuscript presents results of a suite of WRF-LES configurations around the complex topography of the Perdigão field site. Results represent the first time that WRF- LES is used to study the flow interaction in complex terrain and wind turbines. Results of the numerical simulations are qualitatively compared to the experimental measurements. According to the authors, the features of interest for the present work are: (a) mountain waves, (b) recirculation zones, and how these interact with a wind turbine and the corresponding wake. The manuscript is very nicely written, clear, and with superb "candy-to-the-eye" type figures. Therefore one could say that the article is excellent (despite a few typos or unclear elements).

However, to this reviewer's opinion, while the general idea of the work is interesting and filled with interesting challenges and scientific unknowns, the work presented here remains poor in scientific content, and falls short of addressing any of the initial science-based goals. Let me explain, intercomparisons between numerical models and experimental measurements are critical to be able to objectively determine the quality of the simulation results. However, if the effort stops at that, then it only becomes a mere technical work that anyone outside of the academic world could do. There are indeed publications where industry models are intercompared with experimental data, which have been used in the past to provide a sense of confidence or trust to industry. Alternatively, there are also publications from the academic world, where intercomparisons between models and experimental data are done, but in those there is traditionally a critical analysis of the influence of using different numerical schemes, subgrid models, filtering approaches, etc. So at the end there is an evident scientific gain. Unfortunately in the present work the potential scientific gain remains hard to find. What have we learned out of this manuscript? – That we are now entitled to do WRF-LES simulations in complex terrain with turbines? WRF simulations are done on a daily basis around the world, so how will this change what is done on a daily basis? One could have taken advantage of these great simulations to make a more robust objective/quantitative comparison between the simulations and the experimental data. For example, reading that there are wind differences observed at certain times of 1-2 m/s doesn't really mean much. That could

be a 50% difference in a weak mean wind, or a tiny % difference in a strong wind scenario,... Or that the flow looks similar or dissimilar here and there, has little scientific rigor. Once the rigourous/strict comparison of results done, then one could have an additional section with more scientific insight. Research questions that come to mind could be:

As stated in the introduction of the paper, "The focus of this work is to model realistic atmospheric conditions and the associated turbulent flow phenomena to better understand wind turbine wake propagation in complex terrain, using the Perdigão site as the test location. Specifically, we analyze how the vertical deflection and dissipation of the wake varies based on atmospheric stability." This is an open question in the literature based on field observations, and WRF-LES-GAD is now applied for the first time to a real-case simulation with steep slopes to answer the question about wake deflection. The errors in the simulation are quantified via standard metrics (bias and RMSE) compared to a range of field observations. We are cautious to not generalize their results too broadly, but we have attempted to add language that addresses how the results could be interpreted for similar atmospheric conditions and for other sites that may be similar to the Perdigão site (lines 514-518).

1. What is the effect of the surface conditions? – I want to understand that your WRF-LES is using IBM for the topography in regards to the momentum, but is it using the same approach for the ther- mal/moisture field? Based on what I was able to gather from the manuscript it seems that there are no surface conditions for temperature,... does this mean that your flow is insensitive to the surface con- ditions? This is strange since one of the driving mechanisms to thermal stratification is the ground,... but not much is discussed about the impacts or not of the surface conditions on the flow?

The simulations do not use an immersed boundary method - they use the standard terrain-following vertical coordinate system in WRF. We have added language making this more clear in Section 3.3 in line 230. Additionally, the terrain-following coordinate system is mentioned in Table 1. The flow is sensitive to the surface conditions. We use the MYNN surface layer scheme (mentioned in Section 3.3), where the lower boundary conditions are determined from Monin-Obukhov similarity theory, see lines 253-255. The surface temperature is determined based on the Noah land-surface model and a radiative transfer model (the Rapid Radiative Transfer Model, RRTM, stated in Section 3.3). The land-use type is determined using the high-resolution CORINE dataset.

2. What is the effect of your turbulence initialization, and is it really worth it to run a suite of mesoscale WRF simulations just to provide time varying boundary conditions? Mesoscale WRF provides an ensemble flow solution that evolves with time, said otherwise, provides a more or less accurate

mean flow and thermodynamic conditions of the region. However, the LES simulations provide an instantaneous realization of the turbulent ABL, the question then could be posed as how does that compare to instead using profiles of experimental data to force the LES? Also, one can only wonder, what is the effect of the cell perturbation method to generate turbulence? Sure enough the mountains will spur some turbulence, but are the incoming turbulent flow conditions representative of all turbulent scales, including large scale perturbations?

The primary idea behind using multiple grid nests for the simulations is that large scale forcing and perturbations can be passed down to the finer nests where turbulence is resolved. The ultimate goal of such a setup is to use WRF in forecast mode over complex terrain. Although we do not attempt to run LES forecasts in the current work, using observations from the field campaign to force the LES would not achieve progress towards this goal. Furthermore, idealized simulations are unable to provide fully realistic turbulence conditions because the larger forcing scales are not captured (see further discussion in response to specific comment #7 below).

The role of the cell perturbation method has been addressed by Connolly et al. (2021), who studied the effect of using the cell perturbation method versus mountain-generated turbulence in weakly convective, strongly convective, and weakly stable atmospheric conditions for Perdigão. As we mention in line 242-244, they found that the cell perturbation method improves the representation of turbulence relative to the use of high-resolution complex terrain alone. Additionally, Muñoz-Esparza et al. (2014) found that using the cell perturbation method accelerates the generation of turbulence on the inner nest, with no adverse impact on the flow field and negligible computational cost. Also note that the CPM provides perturbations that lead to development of a full range of turbulent scales, as previously presented in several CPM studies, and this detail has been added to the manuscript at line 241.

3. An alternative, potentially the most interesting research question is how can one use the outcome of the rich simulations to understand flow configurations at other locations? Can results be scaled such that the results become generalizable in terms of stratification, mountain slope, terrain complexity? It would be a lot more interesting if the authors used the rich dataset to come up with generalizable relations that enabled one to extract conclusions at other locations without having to run expensive simulations,...

The major research question this article addresses is the vertical deflection of the wake in two distinct atmospheric conditions. Given conflicting literature (Barthelmie et al. 2019 and Menke et al. 2018) regarding the vertical deflection of the wake in different atmospheric stability conditions based on measurements, we use large-eddy simulations to address vertical wake deflection in

complex terrain. The wake deflection has important ramifications for any turbines that could be located downstream.

As previously mentioned, we are cautious to not generalize their results too broadly, but we have attempted to add language that addresses how the results could be interpreted for similar atmospheric conditions and for other sites that may be similar to the Perdigão site (lines 514-518):

"We expect that the conclusions in terms of the wind turbine wake behavior would hold for all convective and stable atmospheric conditions at the Perdigão site as long as the phenomena of interest (recirculation zones and mountain waves) are present. Other phenomena could be modeled using WRF-LES-GAD to examine wind turbine wake behavior in other cases. With regards to other sites, the wind turbine wake behavior would depend largely on the vegetation (surface roughness) and steepness of the terrain."

**More Specific Technical Comments**

1. Line 42; when talking about 'length-scales' one should clarify that these are "turbulent length-scales"

This has been added.

2. Line 53; it is important to clarify the difference in time scales between the LES and the WRF mesoscale simulations. LES provides an instantaneous realization of the flow at high-temporal frequency, while WRF mesoscale provides results representing the outcome of an ensemble of flow marching in time, but that per the ergodicity definition can not be interpreted at the same frequency than otherwise the LES inout/output.

WRF is designed to operate as both a RANS and LES model, and in practice, the output of the model is a time-evolving 3D field, regardless of whether it is in RANS or LES mode. The timesteps in WRF are chosen according to the spatial resolution and stability limits. Multiple grid nests used to transition from RANS to LES are standard practice, e.g. Wiersema et al. (2020), Rai et al. (2017), and Arthur et al. (2020).

3. In line 57; can the authors provide a brief description of the GAD model? Meaning, I don't need to read the reference to find out whether the models is an actuator disk with/out rotation, etc.

We have added the following text to the manuscript at line 59-61: "In the GAD parameterization, thrust and rotational forces computed at the turbine's blades are averaged over a discretized two-dimensional disk formed by their rotation. These forces are then applied to the flow surrounding the turbine."

> 4. In line 82; the authors mention that "the goal of this work is to model realistic atmospheric conditions and the associated turbulent flow phenomena to better understand wind turbine wake propagation in complex terrain". At the end of the manuscript, what have we learned about this that maybe of use as a function of thermal stratification, or in other locations, mountains, terrain, etc?

As mentioned in the introduction (paragraph from lines 64-75), there is conflicting literature regarding the vertical deflection of the wind turbine wake in different atmospheric stability conditions. In this manuscript, we have learned that during stable conditions where a mountain wave occurs, the wake deflects downwards. During convective conditions where a recirculation zone forms, the wake does not mix with the recirculation zone and deflects above it.

Mountain waves occurred for almost 50% of the nights during the intensive observation period (Fernando et al. 2018). For recirculation, Menke et al. 2019 found that reverse flow with wind speeds greater than 0.5 m/s occurred over 50% of the time when the wind direction was perpendicular to the ridges. We have added these details into the manuscript at lines 130-132.

As previously mentioned in this response, these phenomena are site specific and depend on the characteristics of the terrain such as the slope and vegetation (surface roughness). But generally, we would expect that in other places with hilly terrain the wake would deflect up or down depending on stability.

> 5. By the end of section 2.1; what about the surface conditions? If they were not measured (besides roughness) how are they taken care of in the simulations? Is z0,t taken as a fraction of z0?

As mentioned in a response to a previous comment, we use the MYNN surface layer scheme (mentioned in Section 3.3), where the lower boundary conditions are determined from Monin-Obukhov similarity theory. The surface roughness lengths are based on the 100 m CORINE Land Cover dataset and we also use 30 m terrain from SRTM (mentioned in Section 3.3). Aside from the CORINE land cover which is specific to Europe, we have not changed any of the land-surface options which are standard in WRF. For the reviewer's reference, in WRF, z0,t is parameterized as a function of the Zilitinkevich parameter (Zilitinkevich 1995), z0, and Reynolds roughness number.

6. In Figure 2; I could only wonder but what happens to the near surface heat and momentum fluxes? – It is well known in the community the existence of counter gradient heat fluxes, specially in complex terrain. Why not include subpanels with the near surface heat fluxes?

In response to the other reviewer, we have added the Obukhov length to Fig. 2. Shown here is the new Fig. 2 with additional sub panels of the near surface heat and momentum fluxes (included here only for the reviewer - we do not believe the additional panels are needed in the paper). The near surface heat flux during the stable case is around -0.0025 K m/s and 0.025 K m/s during the convective case. During both case studies, the near surface heat fluxes are relatively constant. Comparing the near surface fluxes with large-eddy simulations is complicated above the surface because there is a combination of resolved and subgrid LES quantities. This analysis is the subject of our ongoing work, which focuses on turbulence quantities, but is not included in the present paper.

[Figure]

7. Around line 185; It seems like simulations are initialized with a uniform heat flux. However, it is known also that the heat flux will be rarely uniform in complex terrain. How is that potential effect assessed? Are we left to interpret that the results are indifferent to that surface forcing?

The idealized simulations are indeed initialized with a uniform heat flux because they are formulated as relatively simple studies to inform the setup of the more realistic multi-scale simulation. A recent study showing that the multiscale setup gives better turbulence properties than the idealized setup is presented in Wiersema et al. (2020). This is because idealized turbulence is missing the large scale forcing that results from dynamic downscaling. A sentence in the introduction of the manuscript at lines 54-56 has been edited to mention this:

"Such setups can provide LES with more realistic time-varying inflow conditions directly from the mesoscale simulations, as these setups include the large scale forcing that results from dynamic downscaling capturing a wider range of atmospheric phenomena and more realistic turbulence compared to conventional idealized LES setups (Wiersema et al., 2020)."

In our multi-scale simulation setup, we use topographic shading and a land-surface model to account for non-uniform heat fluxes in complex terrain. This has been added in Section 3.3 from lines 255-256:

"Additionally, topographic shading is enabled to account for shading effects on the surface heat flux in the complex Perdigão terrain."

8. First line in Section 3.3; "Having demonstrated the ability..."; I dare say that to this point not much has been demonstrated besides showing two beautiful figures. Maybe the authors can tone this a bit down.

Our goal with the semi-idealized simulations was to see if the idealized model can capture the deflection of the wake with different phenomena of interest. We have edited the first line in Section 3.3 as follows: "Having demonstrated the ability of WRF-LES-GAD to capture the different types of wake behavior in a semi-idealized setup, we now proceed to the full multi-scale simulation."

9. Line 210; the authors mention that "adequate turbulence is developed", where is this shown? What is the premised/argument used

to judge that? A certain amount of energy at certain scales? A tke scaling of k−5/3? A certain spectral comparison of tke with the field experiment at different heights? – Some of this might be more "objective".

The text in the manuscript has been rephrased to say "spun up for 9 hours". This amount of spin up time is consistent with common practice for WRF. For example, Connolly et al. (2020) used an 8 hour spin-up time, used a 15 hour spin-up time, and Arthur et al. (2020) also used a 9 hour spin-up time.

10.      It is unclear how the Cell Perturbation Method generates different type of turbulence for the stable and unstable stratification.

The cell perturbation method provides perturbations based over the boundary layer depth. The perturbation temperature is drawn from a uniform distribution within a range of potential temperatures that is based on an optimal Eckert number of 0.2 (see Muñoz-Esparza et al. 2015). This is the subject of the studies by Muñoz-Esparza et al. (2018) and Connolly et al. (2020) and more detail regarding the methodology for CPM can be found there and in Muñoz-Esparza et al. (2015).

11.      From line 265 onwards, the text is riddled with subjective comparisons. I would suggest using percentages instead of absolute values when for example comparing wind speeds. Or avoid using comments like:"matches the measurements well", without an actual metric of it. See line 311; "errors are small" in comparison to what?

At lines 311, 317, and 388, subjective comparisons have been removed.

At lines 338, 346-349, 354, and 397-398, when discussing and comparing the errors, we have now quantified the errors with metrics of bias and RMSE within the text.

12.      Why is the wind speed and direction outputted at different frequencies?

The wind speed and wind direction are output at the same frequency, 10 seconds. Perhaps the reviewer is commenting about the temperature gradient output frequency, which was 2.5 minutes, simply because of computational storage limitations.

13. Around line 367; the authors comment that the errors/discrepancies observed during the convective periods are larger than during the stable periods, but they don't provide any comment, hypothesis, argument, detailed inspection, trying to investigate that in details,...

The sentence containing this observation has been removed from the manuscript.

14. Figure17, great visualization for oral presentations, proposals or others where "candy-to-the-eye" is well accepted; but what is the use of such an image here? What is the intended message, outcome extracted? – I am not suggesting the authors remove the figure but instead include scientific argumentation around it.

Figure 17 shows the meandering and full development of the wake over complex terrain. The entirety of the wake cannot be visualized in two dimensions (vertical transects or plan slices) because of wind veer and horizontal/vertical wake meandering. The discussion for Figure 17 is qualitative while Figure 19, later on in Section 4.3, aims to be more quantitative regarding vertical wake behavior. We have added the following sentences in lines 429-431:

"The entirety of the wake cannot be visualized in two-dimensions because of wind veer and horizontal/vertical meandering. Three-dimensional visualizations provide insight into the wind turbine wake advection, meandering, and direction downstream as the flow evolves and develops over the first ridge and through the valley."

In conclusion, I am convinced the data presented in this manuscript is of high quality, and has the potential to become of high value for the community if made publicly available. My only concern is that there is not enough scientific value at presenting the data itself, when additional analysis (some of which does not require extensive work) could be added that would increase the scientific value of the work.

We thank the reviewer for the comments. We think that the revised manuscript and the responses above help to address most of these concerns.

---

## Referee Report (RR1)

**Title:** *Meso- to micro-scale modeling of the atmospheric stability effects in wind turbine wake behavior in complex terrain*

**Authors:** *Adam S Wise, James M T Neher, Robert S Arthur, Jeffrey D Mirocha, Julie K Lundquist, and Fotini K Chow*

**Submitted to:** QJRMS

**Date:** September 1, 2021
* * *
**Reviewer general comment:** *The current manuscript presents results of a suite of WRF-LES configurations around the complex topography of the Perdigão field site. Results represent the first time that WRF-LES is used to study the flow interaction in complex terrain and wind turbines. Results of the numerical simulations are qualitatively compared to the experimental measurements. According to the authors, the features of interest for the present work are: (a) mountain waves, (b) recirculation zones, and how these interact with a wind turbine and the corresponding wake as a function of atmospheric thermal stratification. The manuscript is very nicely written, clear, and with superb "candy-to-the-eye" type figures. Also the a priori aimed research question is of interest, and providing a thorough analysis of the problem posed would be of high interest to the community. Unfortunately the scientific analysis and content remains largely the same as the original version, mostly based on qualitative metrics and comparisons of numbers. The truth is that review of this manuscript has become quite a frustrating experience (hence in part the long delay in the review; my apologies for that), given that the authors instead of carefully considering the constructive suggestions and comments made, have instead taken a defensive approach. Review of scientific manuscripts is a volunteering act in support of the community. I don´t do it with any aim to go against anyone, or their work. I mostly do it to keep learning, and try to provide my opinion for others to consider and stir some additional thinking. As a critical scientist with myself, and with the community, I would reject publication of the manuscript as is. However, I also realize that this might be a controversial decision, hence I rather leave the community to judge it by themselves, and time will tell whether the work herein presented is worthy of publication. Beyond this rather important general comment, below I provide a few additional comments, that if the authors are willing to consider, could probably strengthen their manuscript.*

**More Specific Technical Comments**

1. **Reviewer comment:** In line 9-10; it might be appropriate to try to justify why the authors have selected to two study the mentioned periods. Are they relevant to something, or representative of something? Or is just a random selection. Providing this additional info my help motivate. This also relates to the text in Line 14-15. Are the authors expecting the dependence on the wake behavior to be continuous and smooth, or illustrating a sharp change in behavior at a certain thermal stratification?

2. **Reviewer comment:** In line 18-20; '*This study demonstrates the ability of the ...*'; I wonder has this been questioned in the literature? If the WRF model has been used and validated in a continuous manner through the years, and the GAD model has also been developed and

tested (as mentioned by the authors in line 61 to 63 in the text), where is the need for another comparison? Have the authors introduced any new element in the WRF platform that requires testing and verification? – This doesn't seem to be the case, but maybe I missed it again.

3. **Reviewer comment:** In line 55; '*and more realistic turbulence compared to*'; this is not necessarily true. Why would the turbulence initiation method based on the cell perturbation method provide more accurate turbulence than that on an idealized LES simulation? Maybe, and only maybe, under certain atmospheric conditions where very large scale forcing is of relevance, affecting the near surface turbulence, this could be true. However, this is not necessarily generalizable.

4. **Reviewer comment:** In line 74-75, the objective is nicely stated, which is great. It would be nice if the conclusions came around and provided an answer to this fundamental questions. Unfortunately I have the impression this is not the case, since the study only provides information for two independent stability cases. For example, the authors could also follow up on the results/hypotheses from previous works cited by the authors in line 80-81, and explore self similarity of the turbine wakes, in this case as a function of varying thermal stratification. This would add some interesting generalizable science, beyond the case specific observations.

5. **Reviewer comment:** In lines 87-88, why do the authors want to use idealized LES? – This hardly makes any sense for two reasons: 1. the authors mention from the beginning that they are interested in real conditions; 2. they run the simulations, show two instantaneous snapshots, and don't use that data any further. What is the point? I think the manuscript would be a lot simpler just using directly the realistic WRF-LES platform.

6. **Reviewer comment:** In line 86, '*based on atmospheric stability*'; this statement is a bit generous since the authors only consider 2 different stratifications. This is equivalent to that experimentalist that goes to develop a field experiment and only takes two data values to study a complex problem. One would expect at least three points, to discard the obvious linear fit, isn't it? Maybe saying something about the neutral stratification case to complement? If at the neutral stratification regime results are similar to those observed in convective conditions, what is the intensity on stable stratification needed for the wake to start changing behavior? – These are the kind of research question I would have expected to get resolved or studied in this manuscript as I read the introduction.

7. **Reviewer comment:** In line 161, note that the additional figure sent in the review seems to include periods of counter gradient fluxes (maybe my eyes are betraying me,...), but if this was the case a brief comment could be added, mentioning that those time periods have been discarded for example.

8. **Reviewer comment:** In line 206-207, when talking about the cooling rate, I interpret that being at the surface, is that correct? Mind you it could also be at the top boundary condition. Maybe it is worth clarifying this, is you decide to keep the text about the idealized LES.

9. **Reviewer comment:** In line 209-210, how is the turbulence initialized in the ideal LES cases? To that end, what is the turn over time of the flow around the periodic domain? Does the turbulent flow at least have time to properly develop before reaching the mountain? Does the turbulence reach some sort of convergence prior to reaching the region of interest, meaning the mountains?

10. **Reviewer comment:** In line 219, when discussing the surface roughness. If the authors have access to wind lidar data and three tall met towers, why don't they use the experimental data to extract an approximate z0 value? Regardless of how challenging the experimental data might be, at least the value will be of higher significance than that chosen by the authors (which doesn't seem to have much justification). Unless there is a stronger reason behind that decision that is not explained in the manuscript.

11. **Reviewer comment:** In line 224-225; '*Having demonstrated ..*'; once again, I wouldn't feel comfortable myself making that statement. The authors have only showed two beautiful colored figures showing reasonable results, but there is no strict 'demonstration'.

12. **Reviewer comment:** In line 245; instead of going around a rather lengthy justification of why the method to generate turbulence might or might not work, why don't the authors compare the turbulence spectra with that of the met towers? Wouldn't that be a simple, and really robust way to demonstrate there is no doubt on the approach? If the goal of the paper is to study the effect of thermal stratification in realistic conditions, then what better way to proof that the resultant flow is realistic than comparing turbulence spectra? – This seems to me a fundamental requirement.

13. **Reviewer comment:** In line 269-270, where exactly is the surface roughness being manually modified? It might be interesting to have a figure illustrating the surface roughness used in the different regions of the domain, so if someone else wants to redo these simulations for new purposes can actually do so.

14. **Reviewer comment:** In line 275, why does one need to compare the numerical data with the doppler data '*to understand*'? One can compare numerical data and experimental data to verify the accuracy of the numerical simulations, as it seems to be done in this section, or alternatively one can study in detail one of the datasets, or both at the same time to unravel the physics of the problem, which is not what is done in this section. It might be worth rephrasing this opening sentence to clarify the expectations for the reader.

15. **Reviewer comment:** In the paragraph contained between lines 300 to 305, it is discussed that comparison of averaged data in a non-stationary problem is complicated. This is indeed very much true. In fact, as the authors well indicate, given the random nature of the turbulent flow, comparison of instantaneous snapshots is not very significant. As an alternative, the authors provide some sort of 'cone averaging strategy (data within an easting and northing position)', as well as an additional spanwise averaging within 30m, if understood correctly. While one could discuss whether this approach is significant or not (which is not what I intend here), what at least the authors could do is better justify their decision. Why is this method robust? How sensitive are the results on this averaging strategy? – Given that this is more or less a hand-waved approach, I think it would be nice for the reader to have a better sense of the sensitivity of the outcomes on this analysis.

16. **Reviewer comment:** In lines 368-369; '*the height of the mountain wave does not extend as high in the model compared to the results*'; any thoughts on why this could be? – This kind of physical analysis and interpretation is what could make the manuscript a lot more interesting.

17. **Reviewer comment:** I would strongly encourage the authors to consider shortening the manuscript. While the main goal of the project, as stated in the introduction, is to understand the effect of thermal stratification on wind turbine wakes in complex terrain, the analysis

related to this point only expands between pages 25 to 30, among which only 1.5 pages would be of text and analysis, the rest are figures. So, said otherwise, the main focus and goal of the paper is discussed in 1.5 pages of text out of a manuscript of about 30 pages without counting references. The complementary information provided in the Results section (pages 13 to 25, more than 10 pages) relates to the comparison of the WRF obtained flow with data obtained from the met towers and wind lidars. I have the impression that this is quite out of balance, and maybe the paper should then be framed as a comparison between WRF data and experimental data, which once again I am not so sure on its value given that WRF is used daily around the world, and the manuscript does not introduce any new element or variation of elements in terms of WRF configuration. I think this final page-count also helps illustrates the point I was making in my opening remarks.

---

## Author Response (AR2)

**The authors present a simulation study of the Perdigao field campign, using WRF with 5 nested domains, with the three finest being in LES mode. Comparisons with the field measurements are discussed for two selected cases, i.e. one stable and one convective. Overall, this simulation study can be of interest to the community. However, as it is presented now, I do not believe it should be accepted for publication. Results are poorly discussed, with a lot of hand waving and qualitative comparisons of instantaneous fields, the paper is overly long, and lacks detail in many instances (see further comments below). The author's see the merit of their study mainly in the fact that they are the first to use realistic inflow conditions for LES of the Perdigao field campign, but that is not strictly true, since Wagner et al (Atmos. Chem. Phys, 2019) also presented a nested simulation, albeit with only three levels, with only the finest LES. Given all this, I recommend to thoroughly rewrite this paper, and better analyze results.**

We thank the associate editor for the time spent reviewing our work and for the detailed suggestions and comments. Our study is innovative in that it is the first to nest large-eddy simulations down to 10-m resolution, forced with mesoscale simulations. Further, we use these new simulations to examine turbine wake behavior in highly complex terrain. Current wake models do not account for the flow effects of complex terrain when designing wind farms. At 10-m resolution, we can parameterize the wind turbine rotor and analyze the wake's interaction with turbulent terrain-induced flow features under different stability conditions. To the authors' knowledge, there is no previous study that combines highly complex terrain, various atmospheric stabilities, and a wind turbine rotor parameterization at such high resolution. The study by Wagner et al. 2019 analyzes flow characteristics and low-level jets at a grid resolution of 200 m, over an order of magnitude coarser resolution than used in our work here. The wind turbine's rotor diameter is 80 m and requires a minimum of 10 m grid spacing. We view the merit of our study as an examination of the WRF-LES-GAD configuration's ability to capture both relevant and complicated meteorological and terrain impacts, as well as the impacts of those flow regimes on operating turbines, using the Perdigão site as the test location. These points have been added to the manuscript through responses to specific comments, as detailed below. Note that manuscript has decreased from ~10,000 words to under 8,000 words after these major revisions.

Here is a list of the major revisions made to the manuscript:
- The abstract has been reworded to better frame the motivation and to include justification for the two chosen cases.
- Parts of the introduction have similarly been rewritten to better frame the motivation
- Section 2.2: the Froude number has been defined as the internal Froude number. An average internal Froude number has been calculated and the justification for the two cases has been related to the Froude number.
- The semi-idealized section and any associated discussion have been removed
- Sections 4.1 and 4.2 have been renamed as Validation of the Stable Case Study and Validation of the Convective Case Study, respectively.
    - In these sections, we have interpolated the DLR lidar data onto the same grid as the model, which allowed us to add a third panel to this figure to show the difference between the model and the observations, and additional discussion has been added.
    - We also removed many non-objective statements and tried to be as quantitative as possible.
    - Discussion, in general, has been reduced or removed and this includes comparisons with soundings. Figure 12 (previous submitted manuscript numbering) has also been removed.
- Section 4.3

- Figure 16 (previous submitted manuscript numbering) has been time-averaged and the associated discussion has been modified.
- Figure 19 (previous submitted manuscript numbering) now includes experimental results using DLR and DTU lidars. We have also included error bars for the model results and measurements in this figure.
- We have added a section in the appendix called: Sensitivity of Model Results to Grid Resolution. In this appendix, we discuss how errors and biases for mean flow fields are not necessarily reduced by adding layers but that more fine-scale turbulence is able to be resolved (spectra are shown). We also show time-averaged transects highlighting how recirculation depends on the terrain that is able to be resolved on the grid.

**Comments**

**1. I do not see the point of the semi-idealized runs. It is obvious that this will not work with periodic BCs. There is also not much connection to the rest of the paper. This has probably been a useful internal verification exercise, but is not worthwhile reporting on. I suggest to remove this part.**

Thank you for the suggestion. The semi-idealized section and any associated discussion have been removed.

**2. Discussion is qualitative at best and seems often overly optimistic when reporting own results. For instance: "As seen in Fig. 12, the flow inside the valley and near the two ridges is highly dynamic but very accurately predicted by ..." Such a statement can not be simply based on instantaneous snapshots such as shown in Figure 12. Overall, the paper is very lengthy in discussing a lot of these instantaneous figures (e.g. 6, 7, 11, 12, 16, 17). This part could be cut significantly**

We thank the reviewer for bringing this to our attention. The manuscript's discussion of instantaneous figures has been trimmed significantly. Qualitative discussion in general has been reduced. Specifically (using figure numbers from the previous submission):

- For Figure 6, subpanels (c) and (d) have been removed and a paragraph of discussion has been removed
- The discussion for Figure 7 has been significantly reduced. Additionally, we have added a brief examination on the sensitivity of model results to the uncertainty in the TLS positioning
- Figure 12 and the associated discussion have been removed
- Figure 16 has been time-averaged and the discussion has been modified.

**3. Although there are various mean-field results are shown, I would expect a more careful discussion, and in particular figures showing comparisons between 1h averages simulation and experiment along selected vertical and/or horizontal transects (= a line), including error bars (e.g. using bootstrapping). A side by side comparison of color plots in vertical planes is not really very informative (e.g. Fig. 10, 15, 18). Why are experimental results not added to Figure 19?**

Figures 10 and 15 (now Figures 5 and 9) have been modified. The DLR lidar data have been interpolated onto the same grid as the WRF model output. This allows us to take a difference between the simulations and the measurements. In both Figure 10 and 15 (now Figures 5 and 9), a third sub

panel has been added which shows the difference in the along-transect velocities. Brief discussion on the differences between the simulations and measurements has been added to the manuscript as well.

Figure 19 (now Figure 14) now includes experimental results by taking the difference between the DLR and DTU lidar scans. In this figure, we have also included error bars as +/- 1 standard deviation over the hour used for averaging to provide insight into uncertainty.

**4. Figure 7 (lower panel), Figure 8, Figure 13 – this seems to be the long-time evolution of metmast measurements. How much of this is already contained in the d01 and d02 levels. In general, the authors could do a much better job discussing the added benefit of adding LES layers to the prediction quality of their results. E.g. can you quantify a significant reduction of bias per added layer, etc.**

Thank you for the suggestions. We have added a subsection to the appendix titled "Sensitivity of Model Results to Grid Resolution" which examines the wind speed outputs from the model on d03 (dx = 150 m), d04 (dx = 50 m), and d05 (dx = 10 m) for the convective and stable case studies. We have added figures (Figure A1 and A3) that show time-series and spectra for all three LES domains and the metmast measurements. Tables A1 and A2 look at the biases and RMSE for the time-series for all three domains. Biases and errors are not significantly reduced as the grid resolution increases; however, the resolved wind speed spectra are different with only d05 able to capture similar smaller-scale energy-containing eddies as in the metmast measurements.

The long-time evolution of the met mast measurements are not captured on d01 or d02 because of the coarse grid resolution. With a grid spacing of 6.75 km or 2.25 km for d01 and d02, respectively, the double ridge in the terrain is not resolved. Additionally, the ridge-to-ridge distance is ~1400 m so all three met towers would be within a single cell. However, after we nest to finer grid resolutions, the terrain is more accurately resolved.

Another figure is added, Figure A2, that looks at time-averaged vertical transects of the wind speed across the rotor plane during the convective case. Reverse flow or recirculation is only present on the finest domain, d05, because it is able to resolve the steeper slopes in the terrain. Figure A4 is the same figure but for the stable case and we see that the large scale dynamics do not require a 10 m grid, but the wake and turbulence dynamics do.

**5. Discussion on Brunt-Vaisala frequency and gravity waves throughout the paper is very vague and imprecise. In particular: p6 – there are inversion waves and internal waves. Both are determined by a different Froude number (the first based on the inversion step, the second based on the lapse rate). Both relate to phenomena above the ABL. Please properly discuss and define, incl. height and way in which you measure the Fr numbers, velocity scale, length scale (also e.g. in Figure 3). P 12-13. Where/how do you measure the wavelength of the mountain wave event. This seems to be the horizontal wavelength. P 17. The horizontal wavelength of the wave is not influenced by the Froude number, but forced by the terrain feature, only the vertical wavelength is influenced by Fr number. P. 19 Discrepancy between vertical wavelength in model and measurements seems to be related to the fact that your simulation domain and Rayleigh damping layer is not sufficiently high. Properly discuss and justify your set-up or discuss limitations**

Thank you for the suggestions. To clarify, the Rayleigh damping layer in the multi-scale configuration is for the top 5 km of the domain and the domain height is approximately 20 km. This has been added

into the manuscript at lines 196 and 216. The Rayleigh damping layer may not have been sufficiently high in the semi-idealized configuration; the idealized portion of the manuscript has now been removed at the reviewer's suggestions.

The discussion has been modified significantly. Section 2.2 now reads as follows (lines 124-132):

> "Mountain waves can occur when stably stratified flow approaches a topographic disturbance, such as a mountain ridge. These waves can be described using a ratio of inertial to buoyant forces represented by the internal Froude number (Stull, 1988):
>
> $Fr=\pi U/(WN)$
>
> Where W is the mountain ridge width (defined for the southwest ridge of Perdigão to be 586.5 m on average according to Palma et al. (2020), U is the free stream wind speed and N is the Brunt-Väisälä frequency, defined as:
>
> $N=sqrt(g/\theta \, d\theta/dz)$
>
> Here g is the gravitational constant, $\theta$ is the potential temperature of the environment, and $d\theta/dz$ is the lapse rate or vertical gradient in potential temperature. The internal Froude number can also be defined as the ratio of the natural wavelength of the air ($\lambda = 2\pi U/N$) to the effective wavelength of the mountain ridge (2W)."

Figure 3 (previous submission numbering), which had shown the time-series of the Brunt-Vaisala frequency and Froude number has been removed. Section 2.2.1 reads as follows (lines 164-167):

> "...using the average wind speed (6.3 m/s), temperature gradient or lapse-rate (0.031 K/m), potential temperature of the environment (296.7 K), and width of the southwestern ridge of Perdigão (586.5 m), the average internal Froude number during the period of interest is calculated to be 1.05. With a Froude number close to one, we expect a resonant mountain wave to occur."

Relevant discussion in Section 4.1 reads (lines 239-243):

> "The wavelength of the mountain wave is defined as the distance from the first ridge (where the low-level jet begins to deform) to the first crest of the mountain wave. The model predicts a wavelength of 1220 m (Fig. 4(a)), 13% less than the 1410 m wavelength from the DTU lidars, which is almost exactly the ridge-to-ridge distance of 1400 m (Palma et al., 2020)."

The soundings and any associated discussion has been removed. This includes discussion about the Brunt-Vaisala frequency and Froude number, which has been replaced by the new text above.

**6. page 8: why these particular cases. Better justify**

Previous to Page 8, in Section 2.2, we mention (lines 118-122):

> "Two terrain-induced flow features characteristic of the field site are of interest for the present work: (a) recirculation zones and (b) mountain waves. These flow features may occur at the Perdigão site depending on the time of day (Menke et al., 2019; Fernando et al., 2019; Wagner et al., 2019) and are characteristic of the site, in general. Fernando et al. (2019) found that mountain waves occurred for almost 50% of the nights during the IOP while Menke

et al. (2019) found that recirculation occurred over 50% of the time when the wind direction was perpendicular to the ridges."

These particular cases are chosen because either a recirculation zone or mountain wave is expected to occur and those phenomena were commonly observed during the field campaign. We relate the justification of the cases to the expected behavior of the flow based on the internal Froude number. In Section 2.2.1 (Case 1: Convective Conditions), we have added the following sentence on lines 172-173:

"The negative lapse rate results in an infinite internal Froude number and with wind perpendicular to the ridge a turbulent mountain wake with recirculation should form."

In Section 2.2.2 (Case 2: Stable Conditions), we have added the following sentences on lines 164-167:

"Using the average wind speed (6.3 m s-1), temperature gradient or lapse-rate (0.031 K m-1), potential temperature of the environment (296.7 K), and width of the southwestern ridge of Perdigão (586.5 m), the average internal Froude number during the period of interest is calculated to be 1.05. With a Froude number close to one, we expect a resonant mountain wave to occur."

Additionally, the abstract has been modified to include the following sentence on lines 4-7:

"Two case studies are selected to be representative of typical flow conditions at the site, including the effects of atmospheric stability: a stable case where a mountain wave 5 occurs (as in 50% of the nights observed), and a convective case where a recirculation zone forms in the lee of the ridge with the turbine (as occurred over 50% of the time with upstream winds normal to the ridgeline)"

**7. page 12: why set the surface roughness to 0.5m. This seems rather arbitrary. Did you fit this value to obtain the best results?**

This is explained in lines 225-229 which now read:

"The CORINE dataset seemingly misclassifies the land type in the valley as mixed shrub- land/grassland when the vegetation is mostly tall eucalyptus and fir trees. Likewise, Wagner et al. (2019) concluded that the surface roughness lengths at the Perdigão site based on the CORINE Land Cover data were too small. To account for this, we set the surface roughness length for the mixed shrubland/grassland land use category in the valley to 0.5 m, the same value used in the LES studies of Berg et al. (2017) and Dar et al. (2019)."

**8. page 13, line 290. This is a very qualitative statement. Also in earlier work (e.g. Berg et al), agreement was quite remarkable. Please try to be objective, and focus on verifiable facts**

Thank you for bringing this up. We have removed this sentence.

**9. Eq. 3, 4: better explain that these eqs are only valid for the surface layer. Where, at which height to do you measure them?**

The manuscript states that Eq. 3 and 4 are calculated using data from the tower on the southwest ridge at a height of 10 m (the lowest sensors available). We have added to the manuscript that the equations are only valid for the surface layer. The sentence now reads on lines 151-152:

"The Obukhov length and friction velocity, valid for the surface layer, are calculated at SW_TSE04 using 5 minute statistics from the lowest sensors available at 10 m. "

**I thank the authors for carefully revising the manuscript. All comments raised during the interactive discussion have been taken into consideration and addressed in the author's response. In some places, however, I found that the clarifications stated in the author's response could have been transferred more to the primary manuscript. The information will remain available as the author's response will be published together with the primary manuscript upon final acceptance, so I don't think this calls for another round of reviews. However, I strongly suggest to go over the author's response once more to make sure that all additional information regarding numerical parameters, relevant references, more detailed explanations, reasoning behind certain choices etc. stated in the author's response can also be found in the primary manuscript if it is in any way relevant for the clarity of the manuscript or the reproducibility of the simulations or the postprocessing.**

We thank the reviewer for the time spent reviewing our work and for the detailed suggestions and comments in their first report. From the previous review, we have made major revisions to the manuscript. All of the major revisions are listed below; however, the most relevant changes related to the reviewer's previous response are summarized here:

1. The Froude number, Equation 1, is now defined as the internal Froude number, which is the ratio of the natural wavelength of the air to the effective wavelength of the mountain ridge (From Stull 1988 Chapter 14). This equation is appropriate for the horizontal wavelength and we apologize for misunderstanding the reviewer's previous comments.
2. The semi-idealized section and any associated discussion has been removed, which should alleviate any confusion that there may have been between the two model set ups and configurations.
3. Vertical profiles of 1 h time-averaged along-transect velocity profiles are now included, comparing the model output with lidar data. Error bars +/- 1 standard deviation aim to address uncertainties in the comparison.

Below is a comprehensive list of the major revisions made to the manuscript:
- The abstract has been reworded to better frame the motivation and to include justification for the two chosen cases.
- Parts of the introduction have similarly been rewritten to better frame the motivation
- Section 2.2: the Froude number has been defined as the internal Froude number. An average internal Froude number has been calculated and the justification for the two cases has been related to the Froude number.
- The semi-idealized section and any associated discussion have been removed
- Sections 4.1 and 4.2 have been renamed as Validation of the Stable Case Study and Validation of the Convective Case Study, respectively.
    - In these sections, we have interpolated the DLR lidar data onto the same grid as the model, which allowed us to add a third panel to this figure to show the difference between the model and the observations, and additional discussion has been added.
    - We also removed many non-objective statements and tried to be as quantitative as possible.
    - Discussion, in general, has been reduced or removed and this includes comparisons with soundings. Figure 12 (previous submitted manuscript numbering) has also been removed.
- Section 4.3
    - Figure 16 (previous submitted manuscript numbering) has been time-averaged and the associated discussion has been modified.

- Figure 19 (previous submitted manuscript numbering) now includes experimental results using DLR and DTU lidars. We have also included error bars for the model results and measurements in this figure.
- We have added a section in the appendix called: Sensitivity of Model Results to Grid Resolution. In this appendix, we discuss how errors and biases for mean flow fields are not necessarily reduced by adding layers but that more fine-scale turbulence is able to be resolved (spectra are shown). We also show time-averaged transects highlighting how recirculation depends on the terrain that is able to be resolved on the grid.

**Reviewer general comment:** The current manuscript presents results of a suite of WRF-LES configurations around the complex topography of the Perdig~ao field site. Results represent the first time that WRF-LES is used to study the flow interaction in complex terrain and wind turbines. Results of the numerical simulations are qualitatively compared to the experimental measurements. According to the authors, the features of interest for the present work are: (a) mountain waves, (b) recirculation zones, and how these interact with a wind turbine and the corresponding wake as a function of atmospheric thermal stratification. The manuscript is very nicely written, clear, and with superb "candy-to-the-eye" type figures. Also the a priori aimed research question is of interest, and providing a thorough analysis of the problem posed would be of high interest to the community. Unfortunately the scientific analysis and content remains largely the same as the original version, mostly based on qualitative metrics and comparisons of numbers. The truth is that review of this manuscript has become quite a frustrating experience (hence in part the long delay in the review; my apologies for that), given that the authors instead of carefully considering the constructive suggestions and comments made, have instead taken a defensive approach. Review of scientific manuscripts is a volunteering act in support of the community. I don´t do it with any aim to go against anyone, or their work. I mostly do it to keep learning, and try to provide my opinion for others to consider and stir some additional thinking. As a critical scientist with myself, and with the community, I would reject publication of the manuscript as is. However, I also realize that this might be a controversial decision, hence I rather leave the community to judge it by themselves, and time will tell whether the work herein presented is worthy of publication. Beyond this rather important general comment, below I provide a few additional comments, that if the authors are willing to consider, could probably strengthen their manuscript.

We thank the reviewer for the time spent reviewing our work and for the detailed suggestions and comments on how to strengthen our manuscript.

First, we would like to provide a list of the major revisions made to the manuscript:
- The abstract has been reworded to better frame the motivation and to include justification for the two chosen cases.
- Parts of the introduction have similarly been rewritten to better frame the motivation
- Section 2.2: the Froude number has been defined as the internal Froude number. An average internal Froude number has been calculated and the justification for the two cases has been related to the Froude number.
- The semi-idealized section and any associated discussion have been removed
- Sections 4.1 and 4.2 have been renamed as Validation of the Stable Case Study and Validation of the Convective Case Study, respectively.
    - In these sections, we have interpolated the DLR lidar data onto the same grid as the model, which allowed us to add a third panel to this figure to show the difference between the model and the observations, and additional discussion has been added.
    - We also removed many non-objective statements and tried to be as quantitative as possible.
    - Discussion, in general, has been reduced or removed and this includes comparisons with soundings. Figure 12 (previous submitted manuscript numbering) has also been removed.
- Section 4.3
    - Figure 16 (previous submitted manuscript numbering) has been time-averaged and the associated discussion has been modified.
    - Figure 19 (previous submitted manuscript numbering) now includes experimental results using DLR and DTU lidars. We have also included error bars for the model results and measurements in this figure.
- We have added a section in the appendix called: Sensitivity of Model Results to Grid Resolution. In this appendix, we discuss how errors and biases for mean flow fields are not necessarily reduced by adding layers but that more fine-scale turbulence is able to be resolved

(spectra are shown). We also show time-averaged transects highlighting how recirculation depends on the terrain that is able to be resolved on the grid.

**More Specific Technical Comments**

1. Reviewer comment: In line 9-10; it might be appropriate to try to justify why the authors have selected to two study the mentioned periods. Are they relevant to something, or representative of something? Or is just a random selection. Providing this additional info my help motivate. This also relates to the text in Line 14-15. Are the authors expecting the dependence on the wake behavior to be continuous and smooth, or illustrating a sharp change in behavior at a certain thermal stratification?

We have added in the abstract at lines 4-7 why we have chosen the two selected studies:

"Two case studies are selected to be representative of typical flow conditions at the site, including the effects of atmospheric stability: a stable case where a mountain wave 5 occurs (as in 50% of the nights observed), and a convective case where a recirculation zone forms in the lee of the ridge with the turbine (as occurred over 50% of the time with upstream winds normal to the ridgeline)"

We have also related the justification of the cases to the expected behavior of the flow based on the internal Froude number (Equation 1). In Section 2.2.1 (Case 1: Stable Conditions), we have added the following sentences on lines 164-167:

"Using the average wind speed (6.3 m s-1), temperature gradient or lapse-rate (0.031 K m-1), potential temperature of the environment (296.7 K), and width of the southwestern ridge of Perdigão (586.5 m), the average internal Froude number during the period of interest is calculated to be 1.05. With a Froude number close to one, we expect a resonant mountain wave to occur."

In Section 2.2.2 (Case 2: Convective Conditions), we have added the following sentence on lines 172-173:

"The negative lapse rate results in an infinite internal Froude number and with wind perpendicular to the ridge a turbulent mountain wake with recirculation should form"

We expect the wake behavior to be largely governed by the background flow and ambient turbulence, which in turn depends on the thermal stratification and wind speed. Determining continuous vs abrupt change in wake behavior would require more simulations closer to the transitions. This initial work demonstrates two distinctly different cases that are highly representative of this field site and future work can explore the transition between them. We would expect WRF-LES-GAD to be able to capture the full range of flow conditions in different cases and associated effects on the wind turbine wake.

2. Reviewer comment: In line 18-20; 'This study demonstrates the ability of the ...'; I wonder has this been questioned in the literature? If the WRF model has been used and validated in a continuous manner through the years, and the GAD model has also been developed and tested (as mentioned by the authors in line 61 to 63 in the text), where is the need for another comparison? Have the authors introduced any new element in the WRF platform that requires testing and verification? – This doesn't seem to be the case, but maybe I missed it Again.

The WRF model at the mesoscale has been used and validated in a continuous manner; however, the LES capability is becoming more widely used and is in need of validation, especially in cases where terrain effects are important.. Additionally, WRF-LES-GAD has been previously validated for flat and simple terrain but has yet to be tested for complex terrain.

We motivate our study based based on the following:
- An examination of the WRF-LES-GAD configuration to capture both relevant and complicated meteorological and terrain impacts, as well as the impacts of those flow regimes on operating turbines,
- The literature disagrees on vertical wake deflection in different stability conditions
- Terrain-induced flow phenomena and their effects on turbine wake behavior are not typically accounted for in turbine wake models used for designing wind farms. LES can be used to provide guidance to these lower-fidelity tools.

We feel that this demonstration and validation of the WRF-LES-GAD model for two distinct stability classes shows its utility for simulating wakes over a range of stability conditions. However, because the computational expense of running additional simulations in other stability conditions (e.g. neutral, slightly stable, etc.) is prohibitive, we have chosen to focus on two representative cases that are similar to those commonly observed during the field experiment.

3. Reviewer comment: In line 55; 'and more realistic turbulence compared to'; this is not necessarily true. Why would the turbulence initiation method based on the cell perturbation method provide more accurate turbulence than that on an idealized LES simulation? Maybe, and only maybe, under certain atmospheric conditions where very large scale forcing is of relevance, affecting the near surface turbulence, this could be true. However, this is not necessarily Generalizable.

The quoted sentence does not directly relate to CPM or the turbulence initiation method. Rather, it refers generally to our multi-scale nested setup. To be more clear, the sentence has been rewritten in lines 41-42 as follows:

"Such setups can provide LES with realistic time-varying inflow conditions directly from the mesoscale simulations."

4. Reviewer comment: In line 74-75, the objective is nicely stated, which is great. It would be nice if the conclusions came around and provided an answer to this fundamental questions. Unfortunately I have the impression this is not the case, since the study only provides information for two independent stability cases. For example, the authors could also follow up on the results/hypotheses from previous works cited by the authors in line 80-81, and explore self similarity of the turbine wakes, in this case as a function of varying thermal stratification. This would add some interesting generalizable science, beyond the case specific observations.

As mentioned in the response to the second comment, this demonstration and validation of the model at two distinct stabilities shows that the WRF-LES-GAD model can be useful for simulating wakes over a range of stability conditions. However, the computational expense of these simulations prohibits running a large number of cases. Thus, we chose two cases that aim to be representative of commonly observed conditions. On lines 193-194, we have added the following note:

"the current setup takes roughly eight hours of wall-clock time for five minutes of simulation time on 288 cores"

Additionally, the goal of the work has been reframed and added to the manuscript on lines 78-80:

> "The goal of this work is to examine the ability of the WRF-LES-GAD framework to capture terrain-induced flows and their interaction with an operating wind turbine at the Perdigão site (described further in Sect. 2), to thereby demonstrate the efficacy of mesoscale-to-microscale coupled simulations to improve wind plant simulations in complex terrain."

5. Reviewer comment: In lines 87-88, why do the authors want to use idealized LES? – This hardly makes any sense for two reasons: 1. the authors mention from the beginning that they are interested in real conditions; 2. they run the simulations, show two instantaneous snapshots, and don't use that data any further. What is the point? I think the manuscript would be a lot simpler just using directly the realistic WRF-LES platform.

Thank you for the suggestion. The semi-idealized section and any associated discussion has been removed.

6. Reviewer comment: In line 86, 'based on atmospheric stability'; this statement is a bit generous since the authors only consider 2 different stratifications. This is equivalent to that experimentalist that goes to develop a field experiment and only takes two data values to study a complex problem. One would expect at least three points, to discard the obvious linear fit, isn't it? Maybe saying something about the neutral stratification case to complement? If at the neutral stratification regime results are similar to those observed in convective conditions, what is the intensity on stable stratification needed for the wake to start changing behavior? – These are the kind of research question I would have expected to get resolved or studied in this manuscript as I read the introduction.

While we appreciate the reviewer's desire for a description of how wakes change with atmospheric stability, our work shows two distinct, disparate, complex regimes of behavior in the atmospheric stability regimes experienced at this site. A smooth trend between these two regimes is not necessarily expected. As an analogy, smoke dispersion in a convective boundary layer is fundamentally different from that in a stable boundary layer, and no smooth transition between those regimes is observed. Neither would such a trend be expected here, as we assess the dispersion of a wind turbine wake (which is not as passive as a smoke plume) in an atmosphere influenced by complex terrain. Furthermore, neutral stratification rarely occured at this site during the experimental period (Menke et al. 2020). Specifically, here we analyze how the vertical deflection and dissipation of the wake varies in two distinct but representative atmospheric stability regimes, as discussed in the response to #1 above.

7. Reviewer comment: In line 161, note that the additional figure sent in the review seems to include periods of counter gradient fluxes (maybe my eyes are betraying me,...), but if this was the case a brief comment could be added, mentioning that those time periods have been discarded for example.

We have added the following sentence to the manuscript at line 157-158:

> "The met tower measured periods of counter-gradient heat fluxes during both stable and convective conditions but not during the time periods selected for this tower location."

8. Reviewer comment: In line 206-207, when talking about the cooling rate, I interpret that being at the surface, is that correct? Mind you it could also be at the top boundary condition. Maybe it is worth clarifying this, is you decide to keep the text about the idealized LES.

That is correct; however, we have removed the semi-idealized section.

9. Reviewer comment: In line 209-210, how is the turbulence initialized in the ideal LES cases? To that end, what is the turn over time of the flow around the periodic domain? Does the turbulent flow at least have time to properly develop before reaching the mountain? Does the turbulence reach some sort of convergence prior to reaching the region of interest, meaning the mountains?

These are all great questions but we have removed the semi-idealized section to improve the readability of the manuscript, as suggested above.

10. Reviewer comment: In line 219, when discussing the surface roughness. If the authors have access to wind lidar data and three tall met towers, why don't they use the experimental data to extract an approximate z0 value? Regardless of how challenging the experimental data might be, at least the value will be of higher significance than that chosen by the authors (which doesn't seem to have much justification). Unless there is a stronger reason behind that decision that is not explained in the manuscript.

We appreciate the reviewer's suggestion. We now more thoroughly explained the surface roughness on lines 225-229:

> "The CORINE dataset seemingly misclassifies the land type in the valley as mixed shrub- land/grassland when the vegetation is mostly tall eucalyptus and fir trees. Likewise, Wagner et al. (2019) concluded that the surface roughness lengths at the Perdigão site based on the CORINE Land Cover data were too small. To account for this, we set the surface roughness length for the mixed shrubland/grassland land use category in the valley to 0.5 m, the same value used in the LES studies of Berg et al. (2017) and Dar et al. (2019)."

11. Reviewer comment: In line 224-225; 'Having demonstrated ..'; once again, I wouldn't feel comfortable myself making that statement. The authors have only showed two beautiful colored figures showing reasonable results, but there is no strict 'demonstration'.

This sentence has been removed along with the rest of the semi-idealized section.

12. Reviewer comment: In line 245; instead of going around a rather lengthy justification of why the method to generate turbulence might or might not work, why don't the authors compare the turbulence spectra with that of the met towers? Wouldn't that be a simple, and really robust way to demonstrate there is no doubt on the approach? If the goal of the paper is to study the effect of thermal stratification in realistic conditions, then what better way to proof that the resultant flow is realistic than comparing turbulence spectra? – This seems to me a fundamental requirement.

We have added an Appendix titled "Grid Sensitivity Resolution" which examines the wind speed outputs from the model on d03 (dx = 150 m), d04 (dx = 50 m), and d05 (dx = 10 m). We have added Figures A1 and A3 that show time-series and spectra for all three LES domains and the metmast measurements for both the convective and stable cases.

13. Reviewer comment: In line 269-270, where exactly is the surface roughness being manually

modified? It might be interesting to have a figure illustrating the surface roughness used in the different regions of the domain, so if someone else wants to redo these simulations for new purposes can actually do so.

We appreciate this suggestion to improve the reproducibility of our simulations. Although we have decided not to add a figure, we have added the following text description on line 228 (see response to #10 above):

> "…we set the surface roughness length of the mixed shrubland/grassland land use index in the valley to 0.5 m…"

14. Reviewer comment: In line 275, why does one need to compare the numerical data with the doppler data 'to understand'? One can compare numerical data and experimental data to verify the accuracy of the numerical simulations, as it seems to be done in this section, or alternatively one can study in detail one of the datasets, or both at the same time to unravel the physics of the problem, which is not what is done in this section. It might be worth rephrasing this opening sentence to clarify the expectations for the reader.

We have rephrased the sentence to focus on model validation. The sentence now reads (lines 234-235):

> "The stable case is influenced by a mountain wave event. To validate the accuracy of this event in WRF-LES-GAD, we compare the model with multi-Doppler lidar scans obtained by the DTU lidars."

15. Reviewer comment: In the paragraph contained between lines 300 to 305, it is discussed that comparison of averaged data in a non-stationary problem is complicated. This is indeed very much true. In fact, as the authors well indicate, given the random nature of the turbulent flow, comparison of instantaneous snapshots is not very significant. As an alternative, the authors provide some sort of 'cone averaging strategy (data within an easting and northing position)', as well as an additional spanwise averaging within 30m, if understood correctly. While one could discuss whether this approach is significant or not (which is not what I intend here), what at least the authors could do is better justify their decision. Why is this method robust? How sensitive are the results on this averaging strategy? – Given that this is more or less a hand-waved approach, I think it would be nice for the reader to have a better sense of the sensitivity of the outcomes on this analysis.

We have added the following sentence to the manuscript on lines 260-262:

> "To partially account for this and for any uncertainty of the TLS positioning (estimated to be ±30 m), the wind fields in Fig. 6(a-d) have also been spatially averaged by ±30 m in the span-wise direction."

Additionally, figure 6e (current submission numbering) now includes time-series for an uncertainty of +/- 60 m in addition to +/ 30 m to briefly examine the sensitivity of the model results. We have added the following sentences (lines 265-267):

> "In Fig. 6(e), the predictions for ±30 m and ±60 m are similar because the flow is relatively homogeneous in the spanwise direction (a result of the limited terrain variability in this direction, as seen in Fig. 11 and discussed more in Section 4.3)."

Note that Figure 11 was previously instantaneous and has now been time-averaged.

16. Reviewer comment: In lines 368-369; 'the height of the mountain wave does not extend as high in the model compared to the results'; any thoughts on why this could be? – This kind of physical analysis and interpretation is what could make the manuscript a lot more interesting.

This is likely due to the GFS forcing used as the lateral boundary conditions for the coarsest domain d01. Previous work conducted a sensitivity study on how to force the nested simulations and found that GFS forcing produced more accurate results when compared quantitatively with surface observations. We have thus added "near the surface" to line 220. Additionally, we have added the following sentence to line 247-248:

> "The height of the mountain wave does not extend as high in the model compared to measurements, likely a result of errors in the GFS forcing."

17. Reviewer comment: I would strongly encourage the authors to consider shortening the manuscript. While the main goal of the project, as stated in the introduction, is to understand the effect of thermal stratification on wind turbine wakes in complex terrain, the analysis related to this point only expands between pages 25 to 30, among which only 1.5 pages would be of text and analysis, the rest are figures. So, said otherwise, the main focus and goal of the paper is discussed in 1.5 pages of text out of a manuscript of about 30 pages without counting references. The complementary information provided in the Results section (pages 13 to 25, more than 10 pages) relates to the comparison of the WRF obtained flow with data obtained from the met towers and wind lidars. I have the impression that this is quite out of balance, and maybe the paper should then be framed as a comparison between WRF data and experimental data, which once again I am not so sure on its value given that WRF is used daily around the world, and the manuscript does not introduce any new element or variation of elements in terms of WRF configuration. I think this final page-count also helps illustrates the point I was making in my opening remarks.

Thank you for the suggestions. Again, we would like to highlight that while mesoscale WRF is used daily around the world, its LES capability is not used widely, especially in a multi-scale setup with parameterized wind turbines. Fine-scale terrain-induced dynamic flow features, such as those seen here, require resolving steep terrain and are therefore not represented in mesoscale WRF simulations. Overall, the manuscript has been shortened from close to 10,000 words to under 8,000 words. The complementary information in the Results section has been reframed to focus on validation and has been made more concise. These sections (Sections 4.1 and 4.2) now take up only 6 pages compared to 12 previously.

---

## Author Response (AR3)

Dear Authors,

Thank you very much for the thorough revision of your manuscript after the second round of reviews. I believe your work is now much better justified, and results are much better quantified. Therefore, I'm happy to accept your manuscript for publication. I have added below a few small minor corrections, which I suggest that you resolve during the upcoming publication process.

Sincerely,
Johan Meyers

We thank the associate editor for their work throughout this process and we are pleased that the manuscript has been accepted for publication.

Technical corrections
1. line 24: can be very widely based

The sentence beginning on line 24 has been changed to:

"The dynamics of terrain-induced flow phenomena vary based on the time of the day…"

2. line 43: I would suggest to replace "unique high-fidelity simulation …" with "unique turbulence-resolving simulation …" I know that high-fidelity is used a lot in the community, but in fact this is a bit self-aggrandizing, and not necessarily 100% objective

"Unique high-fidelity simulation" has been changed to "unique turbulence-resolving simulation".

3. Line 45. It is probably correct to also refer to ADM-R of the group of Porte-Agel (BLM 2011) who were, I believe, the first to elaborate this type of approach in detail

Thank you for bringing this to our attention. We have included a citation to Wu and Porté-Agel (BLM 2011) in this sentence.

4. line 121: I suggest to remove the acronym IOP; it is only used once after being defined on line 94

The acronym IOP at line 94 has been removed and line 121 now uses the phrase "intensive operation period".